# Redefining the Cut-Off Ranges for TSH Based on the Clinical Picture, Results of Neuroimaging and Laboratory Tests in Unsupervised Cluster Analysis as Individualized Diagnosis of Early Schizophrenia

**DOI:** 10.3390/jpm12020247

**Published:** 2022-02-09

**Authors:** Natalia Śmierciak, Marta Szwajca, Tadeusz J. Popiela, Amira Bryll, Paulina Karcz, Paulina Donicz, Aleksander Turek, Wirginia Krzyściak, Maciej Pilecki

**Affiliations:** 1Department of Child and Adolescent Psychiatry, Faculty of Medicine, Jagiellonian University Medical College, Kopernika 21a, 31-501 Krakow, Poland; natalia.smierciak@uj.edu.pl (N.Ś.); marta.szwajca@uj.edu.pl (M.S.); paulina.donicz@gmail.com (P.D.); alek.turek@doctoral.uj.edu.pl (A.T.); 2Department of Radiology, Jagiellonian University Medical College, Kopernika 19, 31-501 Krakow, Poland; amira.bryll@uj.edu.pl; 3Department of Electroradiology, Jagiellonian University Medical College, Michałowskiego 12, 31-126 Krakow, Poland; p.karcz@uj.edu.pl; 4Doctoral School of Medical and Health Sciences, Jagiellonian University Medical College, Łazarza 16, 31-530 Krakow, Poland; 5Department of Medical Diagnostics, Jagiellonian University Medical College, Medyczna 9, 30-688 Krakow, Poland

**Keywords:** thyroid, cut-off, clustering method

## Abstract

Thyroid abnormalities, including mild forms of hypothyroidism and hyperthyroidism, are reported as risk factors for the development of a number of neuropsychiatric disorders, including schizophrenia. The diagnostic process still takes into account the extreme ranges of the accepted reference values for serum TSH since the concentration of free thyroxine in the serum does not change by definition. TSH mU/L cut-off values in psychiatric patients are currently clinically considered in the case of extremely high serum TSH levels (>4.0 mU/L). The results obtained in this study suggest that the clinically significant value has a lower TSH cut-off point with an upper limit of 2–2.5 mU/L. The criteria for the differential diagnosis of patients with schizophrenia, however, mainly take into account statutory reference ranges without a background related to the history of thyroid diseases in the family. The results indicate the need to lower the upper cut-off values for TSH among patients with early psychosis, which is related to the potential clinical significance of the obtained values both in the field of clinical evaluation and neuroimaging and laboratory evaluation parameters. The cut-off points obtained with the prior available knowledge coincided with the values established in the unsupervised clustering method, which further confirms the legitimacy of their use in the individualized diagnosis strategy of schizophrenia.

## 1. Introduction

Thyroid hormones (THs) play a key role in the development and proper functioning of the central nervous system (CNS) and influence the susceptibility to mental disorders. Taking into account the interaction between the endocrine and immune systems, THs influence the structure of neurons, microglia, astrocytes and oligodendrocytes, as well as the modulation of pro-inflammatory reactions [1]. Since brain development is strongly dependent on thyroid hormones, any impairment of the dynamics of their release causes neurodevelopmental consequences and irreversible changes in the overall architecture and function of the human brain [2].

The endocrine assessment of the functioning of the thyroid gland is an important element of the differential diagnosis of a patient with the first episode of psychosis or with exacerbation of symptoms [3]. A thyroid storm may lead to potentially fatal hemodynamic and metabolic instability when manifested by acute psychotic disorders [4]. Mental disorders are described in both hypothyroidism and hyperthyroidism disorders [5]. Hypothyroidism is mainly associated with depressive symptoms. Symptoms of an overactive thyroid include anxiety, dysphoria, emotional lability and even manic symptoms. Clinical depression occurs in more than 40% of people suffering from hypothyroidism [6].

Population studies show that the diagnosis of schizophrenia is much more common in patients with hypothyroidism than in those with normal thyroid function [7]. Identifying the pattern of changes associated with thyroid hormones in people with schizophrenia may have important clinical implications. Nevertheless, studies assessing the level of thyroid hormones in people with psychotic disorders have provided mixed results. Only some of the cross-sectional studies provided evidence for a relationship between altered thyroid hormone levels, cognitive deficits and comorbid metabolic syndrome in people with psychotic disorders [8,9,10]. Similarly, the neuroendocrine aspects related to the functioning of the hypothalamic–pituitary–thyroid axis (HPT) in schizophrenia are not well understood, while subclinical or overt depression or bipolar disorder are the most commonly reported atypical symptoms associated with hypothyroidism [11]. Nevertheless, autoimmune hyperthyroidism associated with the production of anti-TPO autoantibodies, which cause activation of this organ, was associated with greater severity of negative symptoms and worse functioning of people in acute psychosis [12], which would indicate the neuroendocrine context of psychosis development. Disorders occurring in thyroid disease at the onset or even in clinically severe states of thyrotoxicosis or hypothyroidism show pathological psychological symptoms that mask or aggravate the underlying disease [13].

Thyrotropin (TSH), a hormone produced by the pituitary gland, is a sensitive marker that detects even slight changes in the levels of thyroid hormones. From a clinical perspective, TSH confirms the diagnosis of hyperthyroidism and hypothyroidism on the basis of the clinical picture, which is quite varied [14]. The upper range of serum TSH concentration above 4–5 mU/L is used as the reference for the clinical diagnosis of hypothyroidism [15]. In the face of technological progress and the use of advanced research tools and methods with increased TSH sensitivity, the described reference ranges are lower, which is also related to the fact that thyroid dysfunction was present in the population of healthy people at lower cut-off points. In over 90% of healthy people, a TSH level below 2.5 mU/L was the most common, while values >2.5 mU/L were associated with the occurrence of Hashimoto’s disease or other abnormalities [16,17,18].

The determination of TSH also allows us to detect even a small degree of deviation in the level of thyroid hormones defined as subclinical hypothyroidism/hyperthyroidism, which is a predictor of the development of overt thyroid dysfunction. However, there is insufficient evidence in the literature on the subject of the relationship between subclinical thyroid dysfunctions and the dynamics of the course of an acute psychotic episode or schizophrenia. There is also insufficient data on the relationship of changes in the functioning of the thyroid gland, in particular the influence of subtle differences in the circulating bioactive glycosylated isoforms of TSH on the course of treatment of people with psychosis [19].

Studies of the thyroid function in schizophrenia make it possible to trace the relationships between the biological and psychosocial risk factors of the disease. Thyroid function is shown to be related to psychosocial factors such as childhood trauma. On the one hand, psychological traumas from childhood are often reported by patients with psychosis [20] and shape both the clinical picture and the effectiveness of the undertaken therapies [21,22,23,24,25,26]. It seems, however, that early life is the so-called time window for the appearance of long-term effects on the endocrine system related to the functioning of the hypothalamic–pituitary–thyroid axis [27,28]. In the literature on the subject, there are no reports explaining these dependencies or mechanisms of the relationship between childhood trauma, its neurobiological correlates, and the functioning of the thyroid gland in the population of people diagnosed with schizophrenia.

The aim of the study was to assess the relationship between the thyroid function, biopsychosocial changes and parameters of the peripheral blood and brain in unsupervised cluster analysis in order to redefine the cut-offs for TSH laboratory ranges as an individualized diagnosis strategy of early schizophrenia.

## 2. Materials and Methods

### 2.1. Participants of the Study

The study involved patients in the course of psychotic decompensation admitted to the Department of Adolescent Psychiatry and the Department of Adult Psychiatry at the Department of Adult, Child and Adolescent Psychiatry at the University Hospital in Krakow after giving informed consent to participate in the study and, in the case of juvenile participants, additional consent from their legal guardians. The participant was able to withdraw consent at any stage of the study. Patients were recruited after the approval of the study protocol issued by the Bioethics Committee of the Jagiellonian University (consent number: 122.6120.23.2016 of 23 June 2016 and 1072.6120.152.2019 of 27 June 2019). The subjects were aged 14–35 years and diagnosed with acute multiform psychotic disorders (F23) according to version 10 of the International Statistical Classification of Diseases and Related Health Problems (ICD-10) [29]. The diagnosis was confirmed at initial assessment by two independent psychiatrists. All subjects met the criteria of F20 according to ICD-10, which was confirmed during a 3-month follow-up.

The exclusion criteria were: inability to express informed consent, intellectual disability, hospitalization without the patient’s consent, substance abuse or smoking within three months before admission, affective symptoms, previous head injuries with loss of consciousness, alcohol addiction, hyperactivity or agitation or psychomotor problems that make it difficult to perform magnetic resonance imaging (MRI). Criteria for exclusion from the study also included autoimmune diseases, acute inflammatory diseases, active or previous oncological diseases, chronic terminal disease, cardiovascular disorders, history of thyroid dysfunction (including a history of thyroid surgery or metabolic radiotherapy with radioactive iodine), diabetes and history of CNS disorders. In addition, the study excluded patients taking drugs that may affect the concentration of thyroid hormones and their precursors: oral contraceptives, estrogens, phenytoin, anti-thyroid drugs (e.g., propylthiouracil), lithium, propranolol, glucocorticoids, mineralocorticoids, anti-epileptic drugs, non-steroidal anti-inflammatory drugs and other immunomodulation therapies.

Ultimately, 40 patients were included in the study, i.e., 18 women and 22 men (mean age 22.68 ± 7.39 years). Demographic and clinical data were collected from each patient; they were also asked to complete questionnaire surveys.

In the first week of hospitalization after medical stabilization was achieved, routine blood tests and clinical assessment were performed based on additional psychiatric and psychological examinations. A neuroimaging study was also performed. The imaging tests included MRI of the brain (T1W, T2W, FLAIR, DWI, SWI sequences) to assess possible pathologies in specific regions of the brain; and 1H-MRS studies to quantify and qualify the spectrum of neurometabolites (i.e., myoinositol) selected on the basis of preliminary pilot studies [30]. The field strength of 1.5 Tesla was used in the conducted research. MRI and MRS were performed during the first two weeks of hospitalization in patients with no changes in outpatient status and those not receiving pharmacotherapy 8 h before each brain imaging.

The study participants were using pharmacotherapy at the time of the study. Both classical neuroleptics and second-generation neuroleptics were used in accordance with the guidelines of the American Psychiatric Association [31]. In the group of patients in the first week of clinical observation, the following anti-psychotic drugs were used: haloperidol, including one with aripiprazole, olanzapine, risperidone and quetiapine. At the second time point, patients were taking quetiapine, aripiprazole and olanzapine. Due to the used combination therapy, the studied group of patients included three patients treated with partial agonists of the D2 receptor (mainly aripiprazole in the drug group) in the first week of observation, with one patient using it as monotherapy, one combined with haloperidol and in the second case with quetiapine. However, after 12 weeks of observation of treatment with partial agonists of the D2 receptor (i.e., mainly with aripiprazole), 10 patients were treated with two drugs. Drug doses were converted to chlorpromazine equivalents. Chlorpromazine equivalent (CPZE) is defined as a drug dose that corresponds to 100 mg of an oral dose of chlorpromazine [32]. A drug equivalent to 200–300 mg of chlorpromazine is considered a minimally effective dose, while more than 1000 mg of chlorpromazine is considered high [33]. None of the patients experienced side effects.

### 2.2. Research Procedures

The study used a number of research tools relating to clinical conditions, emotional functioning of the patient, measurable routine and specialized indicators of imaging, laboratory assessment, as well as risk factors. The doses of neuroleptics used were converted into chlorpromazine equivalents [34]. These variables were used in the study. The description of the used research methods is included in Table 1.

The assessment of psychotic symptoms was performed using The Positive and Negative Syndrome Scale (PANSS) [35]. The PANSS scale is a 30-point tool for assessing the severity of schizophrenia symptoms. To assess the patient’s condition by PANSS, an interview of approximately 45 min with a clinically experienced psychiatrist was conducted with each patient. Items were rated on a scale of 1 (asymptomatic) to 7 (extremely symptomatic). The conducted analyses were based on the five-factor model, which better characterizes the results from the PANSS scale in patients with schizophrenia than the traditional three-factor model [45,46,47,48]. The scales included factors for positive symptoms (PANSS pos), negative symptoms (PANSS neg), disorganized thoughts (PANSS dis), uncontrolled hostility/excitement (PANSS exc) and anxiety/depression (PANSS emo) [47]. The individual factors include the following items from PANSS: Positive symptoms: delusions (P1), hallucinatory behaviour (P3), grandiosity (P5), suspiciousness (P6), stereotyped thinking (N7), somatic concern (G1), unusual thought content (G9), lack of judgment and insight (G12);Negative symptoms: blunted affect (N1), emotional withdrawal (N2), poor rapport (N3), passive social withdrawal (N4), lack of spontaneity (N6), motor retardation (G7), active social avoidance (G16);Disorganized thoughts: difficulty in abstract thinking (N5), mannerisms and posturing (G5), disorientation (G10), poor attention (G11), disturbance of volition (G13), preoccupation (G15), conceptual disorganization (P2);Uncontrolled hostility/excitement: poor impulse control (G14), excitement (P4), hostility (P7), uncooperativeness (G8);Anxiety/depression: anxiety (G2), guilt feelings (G3), tension (G4), depression (G6) [47].

Depressive symptoms were measured using the second edition of The Beck Depression Inventory (BDI-II) [36]. BDI-II is a widely used self-report tool for measuring the severity of depression symptoms in psychiatrically diagnosed patients. It consists of 21 items, and its total score is from 0 to 63. Higher scores indicate greater severity of depression. The cut-off points are: 0–13: minimal depression; 14–19: mild depression; 20–28: moderate depression; 29–63: severe depression. The Polish adaptation was used in the study [49]. Additionally, The Calgary Depression Scale for Schizophrenia (CDSS) was used [37]. CDSS is a widely used tool to assess the level of depression in schizophrenia. The survey consists of nine items, rated from 0 (asymptomatic) to 3 (extremely symptomatic). The overall range of scores ranges from 0 to 27 points. The higher the overall score, the greater the severity of depressive symptoms [37,38]. A score above 6 has an 82% specificity and 85% sensitivity for predicting the presence of a major depressive episode. It is shown that the scale measures depression and not negative or extrapyramidal symptoms [50].

The presence of traumatic symptoms in the past was assessed on the basis of the Childhood Trauma Questionnaire (CTQ), which is a retrospective self-assessment tool providing a description of trauma experienced in childhood and adolescence [39,40]. It consists of 28 items; each item of the CTQ is rated on a five-point scale with response options ranging from ‘never true’ to ‘very often true’. CTQ measures five types of trauma: emotional (CTQ-EA), physical (CTQ-PA), sexual (CTQ-SA), emotional (CTQ-EN), physical neglect (CTQ-PN), and gives a total score (CTQ-Total). The total score is the sum of the five subscales (CTQ-EA + CTQ-PA + CTQ-SA + CTQ-EN + CTQ-PN). The score for each type of abuse ranges from 5 to 25 points. EA refers to verbal assaults on a child’s sense of worth or wellbeing or any humiliating, demeaning or threatening behavior directed toward a child by an older person. PA refers to bodily assaults on a child by an older person that poses a risk of, or results in injury. SA refers to sexual contact or conduct between a child and an older person, including explicit coercion. EN is characterized by the failure of caregivers to meet the child’s basic psychological and emotional needs, such as love, belonging and support. PN is a failure of caregivers to provide a child’s basic physical needs. The CTQ can be considered the most frequently used scale to assess child abuse in the literature [1]. In this study, the Polish version of the CTQ was used [51].

The PANSS, BDI-II and CDSS scales were used at two time points: during the first week of hospitalization (_1) and three months after admission to the hospital (_12). CTQ was performed on patients when clinically stabilized within 3 months of admission.

### 2.3. Description of Biochemical and Neuroimaging Analyses

#### 2.3.1. Laboratory Analyses

Blood samples were collected from the enrolled fasting patients between 6–8 a.m. Routine blood laboratory tests included blood count (5 diff), lipid profile (low- and high-density lipoproteins, LDL and HDL; triglycerides, TG; and total cholesterol, TC), inflammatory markers (C-reactive protein, CRP; ESR), ionogram (K^+^, Na^+^, Mg^2+^), glucose, creatinine, urea, γ-glutamyltransferase (GGTP) and tests assessing the functioning of the thyroid gland (free triiodothyronine FT3, free thyroxin FT4, and thyroid-stimulating hormone TSH, released by the pituitary gland and stimulating thyroid for the production of FT3 and FT4). Routine analyses were performed on the day of blood collection in the laboratory of the University Hospital in Krakow using XN-2000 automated analyzers (Sysmex Corp., Kobe, Japan) as well as Cobas 6000 and Cobas 8000 biochemical analyzers (Roche Holding AG, Basel, Switzerland). The laboratory of the University Hospital in Krakow is subject to daily internal quality control (Precicontrol ClinChem Multi 1, P Precicontrol ClinChem Multi 2, Lyphochek Assayed Chemistry Control level 1, Lyphochek Assayed Chemistry Control level 2, Precicontrol Universal level 1 and Precicontrol Universal level 2 for biochemical parameters; Sysmex XN-Check at three levels) and systematic external quality control (at the Central Center for Quality Research in Laboratory Diagnostics, Poland and the Randox International Quality Assessment Scheme), in accordance with the accepted standards for medical diagnostic laboratories [52].

#### 2.3.2. Specialized Laboratory Analyses

##### Total Antioxidant Potential Expressed as FRAP in Blood and Saliva

The FRAP measurement provides information about the antioxidant capacity of plasma and saliva to counteract the effects of free oxygen radicals. 

Stimulated saliva samples were collected from all participants using the Salivette^®^ Cotton Swab system (Sarstedt AG & amp; Co., Numbrecht, Germany). The material was collected during fasting after rinsing the mouth with distilled water for 30 s. A sterile cotton swab was inserted into the mouth and chewed according to the manufacturer’s instructions for 3 min, thus stimulating the secretion of saliva. Subsequently, the swab soaked in saliva was centrifuged at 1000× *g* for 2 min at 4 °C. The homogeneous, lint-free filtrate was transferred to sterile 0.2 mL Eppendorf microtubes and frozen at −80 °C until the time of analysis.

The concentration of FRAP was determined according to the method of Benzie and Strein [44,53]. Samples of saliva and serum (in appropriate dilution) in a volume of 15 μL were mixed with 300 μL of the reaction mixture containing: 100 mL acetate buffer (pH 3.6), 10 mL 5 mM tripyridyltriazine dissolved in 40 mM HCl and 10 mL 20 mM ferric chloride. The FRAP concentration was calculated by preparing an aqueous solution with a known concentration of Fe II (FeSO_4_⋯7H_2_O) in the range of 0–1.0 mM and a blank sample containing a mixture of FRAP reagents with distilled water. The absorbance of the tested, standard and blank samples was assessed at a wavelength of *λmax* = 593 nm (FLUOstar Omega spectrophotometer; BMG Labtech, Ortenberg, Germany). FRAP values were expressed in mM Fe^2+^/mg protein. Protein concentration in the test material was measured by the bicinchoninic acid (BCA) method according to the manufacturer’s instructions (Sigma-Aldrich, 3300 S 2nd St #3306, St. Louis, MO 63118, USA). The BCA method is based on the reduction of Cu^2+^ to Cu^+^, and Cu^+^ ions react in an alkaline environment with BCA, which gives it a purple color. Absorbance readings were conducted at a wavelength *λmax* = 562 nm at 37 °C (FLUOstar Omega spectrophotometer; BMG Labtech). Protein concentration was calculated from the simple regression equation.

##### Measurement of the Activity of Paraoxonase 1 (PON-1) in the Blood

Plasma PON-1 activity was determined by the method described by Eckerson et al. [41], as modified by the authors. The absorbance of the resulting p-nitrophenol was recorded spectrophotometrically at *λmax* = 405 nm at 25 °C using a FLUOstar Omega microplate reader (BMG Labtech, Ortenberg, Germany). The plasma samples were mixed with a buffer containing 1.2 mM paraoxone in a 50 mM glycine buffer containing 1 mM CaCl_2_ (pH 10.5). The samples were then incubated for 15 min at 37 °C. In the next step, 20 µL diluted plasma was measured, 200 µL of 1.2 mM paraoxone was added and the absorbance value was monitored at *λmax* = 405 nm every 15 s for 4 min after being gently mixed.

The results are expressed in international units (U/L): PON-1 = OD/min × 11.4 = U/L.

##### Measurement of the Severity of Lipid Peroxidation by Assessing Blood Malondialdehyde (MDA)

Strong oxidative stress damages cell membranes and is an important element in the pathophysiology of schizophrenia [54]. Malondialdehyde or malonyldialdehyde (MDA) is a product of lipid peroxidation, the production of which is reflected in the breakdown of polyunsaturated fatty acids (PUFA).

The quantification of MDA is a good indicator of the degree of severity of lipid peroxidation in vivo. The quantification is possible by reacting MDA with thiobarbituric acid (TBA) and measuring the produced pink MDA-TBA adduct. In the presented method of fluorimetric TBA-MDA adduct determination, we observe unique absorbance maxima at *λmax* = 535 nm, fluorescence emission at *λmax* = 553 nm, and excitation at *λmax* = 532 nm. The formation of the TBA-MDA condensation product is carried out with the addition of BHT according to the method described by Aust [42] with Gutteridge’s modification [55].

The working solution was prepared by diluting the standard solution containing 2-thiobarbituric acid, TBA, trichloroacetic acid (TCA), and HCl in water (prepared on the day of the determination), thus obtaining 3.5 mL of TBA/TCA/HCl + 10.5 mL H_2_O + 0.21 mL BHT. 1,1,3,3-tetramethoxypropane was used as a standard, which hydrolyzes in acid in a stoichiometric ratio to MDA. Hydrolysis was carried out in 0.05 M HCl at room temperature for 10 min, then standard solutions of 1,1,3,3-tetramethoxypropane were prepared in the concentration range: 0.015–3.5 µM. Before starting the proper determinations, two standard concentration ranges were checked in accordance with the MDA levels presented in the literature: 0.015–10 µM and 0.015–3.5 µM. A lower range of standard concentrations was chosen for the experiment, which resulted from the observed MDA levels in the test and control groups.

The working solution was prepared daily fresh from TBA/TCA/HCl solution by dissolving in water in a ratio of 1:3. Stock is stable for several months if stored under proper conditions (room temperature).

Test serum, blank or reference samples were mixed with the working solution in the ratio: 125 µL sample and 1000 µL working solution. The contents of the test tubes were mixed for 10 s with the use of a micro-shaker and then heated in a boiling water bath for 15 min. The tubes were then immediately chilled on ice for 10 min, and 3 mL of butanol was added to each tube. The reaction mixtures were shaken for 30 s. After centrifugation for 10 min at 4000× *g* at room temperature, 250 µL organic layer was carefully transferred to the wells of a black 96-well plate. Fluorimetric measurements were conducted at an excitation wavelength (Ex) of 536 nm and an emission wavelength (Em) of 549 nm. The readings were conducted after 10 min using the FLUOstar Omega spectrophotometer (BMG Labtech, Ortenberg, Germany).

#### 2.3.3. MRI

Magnetic resonance imaging (MRI) was implemented using the 1.5 T magnetic field induction MR system eight-channel head coil in a horizontal position (General Electric Healthcare, Milwaukee, WI, USA). The project was organized using a standard MR brain examination protocol including sequences: FLAIR, T1-weighted, T2-weighted and diffusion-weighted imaging (DWI). Highly repeatable, fast and accurate scans could be developed by equipping the MR system with firm whole-body gradients providing an amplitude of 33 mT/m and a rising rate of 120 T/m/s on each axis. Sagittal, axial and coronal two-dimensional T2-weighted images were achieved to show the plan position of the volume of interest (VOI), (rectangle) and anatomical brain structures for spectroscopy [56]. Magnetic resonance spectroscopy (MRS) was executed using the single-voxel technique (SVS). Acquisition of MRS spectra was possible using a point-resolved spectroscopy sequence (PRESS Point—Resolved Spectroscopy Sequence). Radiofrequency pulses: one 90° and two 180° were operated by PRESS sequence. The CHESS sequence (CHEmical shift Selective Imaging Sequence) was implemented for water suppression with a frequency-selective 90° pulse and dephasing gradient to destroy the water signal. The CHESS technique is based on the selective attenuation of the signal from solvent molecules prior to non-selective excitation of the entire frequency range. This requires the use of an RF pulse operating only in the 50 Hz frequency range with the center frequency in the water peak. The RF pulse tilts the magnetization from the water to the transverse plane, where the magnetization is then de-phased by a gradient pulse, the so-called crusher. The then generated sequence of non-selective RF pulses already excites the entire frequency range, but the saturated solvent signal is suppressed. In the tests performed, the water line was saturated with three selective rf pulses, which made it possible to reduce its amplitude 103–104 times. Automatic shimming was used to achieve satisfying spectra. This is an automatic procedure to improve the homogeneity of the magnetic field after a patient is placed in it and applies to the volume of the selected voxel (voxel shimming). Due to the fact that the body of each patient influences the magnetic field slightly, shimming allows for local improvement of the field uniformity and, as a result, the quality of the spectrum. The shim coils are responsible for maintaining the best homogeneity of the magnetic field. The measurement of homogeneity is facilitated by measuring the width of the water peak at half its height. This is called half-width w1/2. For high-resolution spectroscopy, it should be less than 0.2 ppm (by default, it is about 0.1 ppm). MRS acquisition specifications of 35 ms TE with 64 averages were obtained. During the project, MRS signal was obtained from three locations situated symmetrically in the anterior cingulate cortex (ACC), right and left frontal lobes, centered on the interhemispheric fissure and parallel and superior to the dorsal anterior surface of the corpus callosum. VOI reached approximately 8 cm^3^. The extent of the VOI was adapted to the anatomical size of the location the spectrum was obtained from. The period of the sequence lasted 2 min 12 s. 

After the test was performed, signals from specific VOIs were obtained in the form of Free Induction Decay (FID) signals. Their mathematical counterparts were numerical data files containing basic (raw) data. They were then processed for:presentation of data in an easy-to-interpret form;assigning specific signals to specific metabolites;determination of absolute and relative concentrations of the metabolites.

Analysis of the MRS data was performed using SAGE 7.0 (Spectroscopy Analysis by GE). The analysis was carried out in the following stages:loading of appropriate raw data presented in the form of FID signal;data reconstruction (reconstruct-Probe Quant), which consists of noise reduction by applying filters and automatic Fourier transformation (conversion of a time-domain function to a frequency-domain function);offset correction—subtraction of the signal component that arose in the receiver as a result of the interaction of electronic circuits, while the FID signal disappeared to zero;zero filling—supplementing the digital form of the signal with additional data of zero amplitude in order to improve the resolution of the spectrum;apodization—signal multiplication by appropriate mathematical functions, which improves the signal-to-noise ratio;determination of peaks that were subject to further analysis;converting the spectrum with marked peaks into the FID signal (processing FID generate);Fourier transformation of the newly received FID signal;reading the value of the area of the fields under the selected peaks;superimposing the developed spectrum with the selected peaks on the original spectrum;calculation of relative ratios of metabolite concentrations in relation to Cr.

Metabolites were manually chosen from the spectrum: lipids (lip 0.9–1.0 ppm), lactates (lac 1.33 ppm), alanine (ala 1.8), N-acetyl-aspartate (NAA 2.02 ppm), glutamate (glu 2.1 ppm), γ-aminobutyric acid (GABA 2.3 ppm), glutamine (gln 2.45 ppm), creatine (Cr 3.02 ppm), choline (Cho 3.22 ppm), glucose (glc 3.43 and 3.8 ppm), myoinositol (mI 3.56 and 4.0 ppm), glu + gln + GSH complex (3.7 ppm) and phosphocreatine (3.9 ppm).

### 2.4. Statistical Analysis

Statistical analysis was performed using the IBM SPSS Statistics 25 package. The Spearman correlation analysis was used, which allowed us to check whether there is a statistically significant relationship between the analyzed variables. Analysis with the Mann–Whitney U test made it possible to check whether there were statistically significant differences between the adopted cut-off values separated on the basis of the TSH level. An unsupervised cluster analysis was used to assess the relationship between all recorded TSH values and the clinical and laboratory assessment of patients in early psychosis. The individual laboratory parameters that were tested in this article were entered into the cluster analysis. In the case of introducing different parameters at the same time, the standardization of the variables was carried out for this purpose. TSH itself was not included. However, after carrying out these analyzes, it turned out that no additional parameters were observed that, when included in the analysis with TSH, would significantly change the distribution of results. The parameter that allowed to differentiate the examined patients to the greatest extent was TSH. It was the only cluster analysis that allowed for the identification of equal groups of patients. Using the k-means cluster analysis (it should be noted that the k-means approach works by minimizing intra-cluster variance), two clusters of people were distinguished, i.e., with the maximum possible difference in the level of TSH. 

The following descriptive statistics were used in the analysis: mean, median, standard deviation, minimum, maximum, first and third quartile. The *p*-value <0.05 was adopted as statistically significant.

## 3. Results

### 3.1. Relationship between the Functioning of the HPT Axis and the Analyzed Variables

In the first step, it was checked whether there was a statistically significant relationship between the parameters related to the functioning of the thyroid gland, clinical assessment related to the course of the disease and the severity of symptoms, the declaration of the occurrence of traumatic experiences and laboratory parameters, both routinely performed (complete blood count, biochemical tests) and specialized (apart from standard research protocol, i.e., neuroimaging of the diffusion tensor in the frontotemporal region, as well as biochemical methods assessing the severity of oxidative stress) (Table 2, Table 3, Table 4 and Table 5). The relationships between the functioning of the thyroid gland, clinical parameters, imaging parameters, i.e., fractional anisotropy (FA) and metabolic parameters, were analyzed using general linear models and cluster analysis without prior available knowledge.

Among the examined oxidant-antioxidant parameters, two statistically significant relationships were observed, namely FT3 with MDA µmol/L_12 and PON1 U/L_12. The higher the FT3 level, the higher the value for PON1 U/L_12 and the lower in the MDA µmol/L_12 range (Figure 1). In the case of inflammatory indicators, the relationship between CRP levels and positive symptoms was also investigated. The occurrence of two statistically significant correlations were observed, i.e., CRP_1, with newP1_1, rs = 0.34; *p* = 0.04 and pos1, rs = 0.34; *p* = 0.03. This means the higher the level of CRP_1, the higher the score of these two variables. These results, however, were the subject of our earlier research [57]. According to the obtained results in young patients with the first psychotic episode of FEP, pro-inflammatory reactions are caused by a severe onset of acute psychosis and should be treated as systemic symptoms of FEP.

### 3.2. Analysis of the Tested Parameters of Clinical and Laboratory Evaluation According to the Level of TSH

In the constructed model of a number of variables, the relationship between the functioning of the thyroid gland and the level of TSH was assessed for potential clinical implications or routine and specialist parameters (both laboratory and neuroimaging), taking into account the cut-off points (1) 0–2.5 mU/L and (2) >2.5 mU/L.

Two groups of subjects were distinguished, i.e., according to the level of TSH, namely 1–2.5 mU/L (n = 20, 54.1%) and 2.5–4.2 mU/L (n = 17, 45.9%). Table 6, Table 7 and Table 8 present descriptive statistics on the analyzed variables divided into two separate groups of patients.

Statistically significant differences that were observed between the two separate groups of people concern the following variables: PANSS_neg_12, BDI_12, Calgary_12 and LYMPH × 10^3^/µL_1. A statistically significantly higher result for people with TSH levels 0–2.5 mU/L applies to the following variables: PANSS_neg_12, BDI_12, Calgary_12. The opposite situation applies to LYMPH × 10^3^/µL_1. The greatest differences are characteristic of BDI_12 (Figure 2). 

### 3.3. Analysis of the Relationship between the Functioning of the Thyroid Gland and Individual Variables

In the case of TSH level, there are several statistically significant relationships, two of which are noteworthy, i.e., with BDI_12. The higher the TSH level, the lower the value for both variables. The strongest relationship concerns the BDI_12. Among the remaining TSH level relationships, two negative ones, i.e., with FA right frontal lobe DEV and ACC_MI, are significant. A stronger relationship is noted in the case of FA right frontal lobe DEV.

In the case of FT4, two relationships are noteworthy, namely with CTQ_EA and CTQ_PN. The higher the value of the FT4 level, the higher the value for these variables. A stronger relationship is in the case of CTQ_EA. Among others, the relationship of FT4 with ACC_MICR is also noteworthy.

In the case of FT3, relationships with the number of hospitalizations neg_1 and pos_12 are significant. The strongest relationship is with neg_1. The associations with FRAPsaliva_1 (positive) [58], FA_ACC left AVG (negative), ACC_GLUCR (positive) and ACC_MICR (positive) are noteworthy. The strongest relationship also applies to ACC_MICR.

The graphic description of the abovementioned relationships is presented in Figure 3.

### 3.4. Cluster Analysis

In the next step, a cluster analysis was performed. Two groups of patients were distinguished, i.e., divided according to the level of TSH, and then it was checked for which variable they differ in the greatest possible way. The variables that differentiate the selected clusters to the greatest extent and, at the same time, have an impact on the quality of life of the respondents were searched successively (the fulfilment of two conditions is more important in practice).

One very important issue was highlighted: the analysis of clusters without supervision distinguished two clusters of people on the basis of the TSH level, which is identical to the division with prior available knowledge; TSH level (1–2.5 mU/L and 2.51–4.2 mU/L). Descriptive statistics on the analyzed parameters are presented in Table 9.

The selected clusters show identical descriptive statistics of the analyzed variables, as in the division into groups of people based on the TSH level (1–2.5 mU/L; 2.51–4.2 mU/L). Thus, the analyses were not repeated. Statistically significant differences that were observed between the two groups of people concern the following variables: PANSS_neg_12, BDI_12, Calgary_12 and LYMPH × 10^3^/µL_1. A statistically significantly higher result for people with TSH levels from 1 to 2.5 mU/l applies to the following variables: PANSS_neg_12, BDI_12, Calgary_12. The opposite situation applies to LYMPH × 10^3^/µL_1. The biggest differences are characteristic of BDI_12.

Thus, when TSH is tested, its greatest impact in terms of mental state is related to the variables listed above. One can name patients with TSH as, e.g., cluster 1, cluster 2, or group 1 and group 2 based on TSH level (1–2.5 mU/L; 2.51–4.2 mU/L). Comparisons between the separated clusters do not have to be made because they are identical, which corresponds to the cut-off points prevailing in science and corresponding to the border of the so-called clinical horizon.

No direct relationship was observed between the results of thyroid parameters and the intensity of pharmacotherapy measured by chlorpromazine equivalents (Table 10).

The individual TSH compounds are weaker; however, the relationship with BDI_12 is noteworthy. A ROC curve was created for this parameter. The lower the TSH level, the higher the BDI_12 score. The obtained score was divided into two groups of patients: up to 15 points and above according to the adopted concept [59,60]. The TSH level was tested once, and the BDI was measured after 12 weeks of clinical observation. The ROC curve is related to the prediction of BDI_12 from the baseline TSH concentration (Figure 4). Baseline TSH_1 showed no such result with baseline BDI, but after 12 weeks of the study.

Noteworthy is the fact that the differences also concerned the following variables: PANSS_neg_12, BDI_12 and Calgary_12, but the greatest differences are characteristic of BDI_12.

## 4. Discussion

This study presents the changes in the clinical picture and brain activity in connection with the functioning of the HPT axis. Deviations from the norm in the group of patients with schizophrenia are widely described in the literature. According to a meta-analysis from 2021, an increased level of TSH was observed in patients with multiple psychotic episodes in relation to healthy people, while in patients in the first psychosis, a reduction in TSH and FT3 levels and an increase in FT4 levels compared to the control group were observed [61]. The authors of the quoted study point out the need to understand the mechanisms of HPT axis deviations in patients with schizophrenia. Due to the longitudinal nature of our research and the use of advanced neuroimaging studies, our study allows us to locate potential mechanisms that may be involved in the interaction between deviations within the HPT axis and the broadly understood clinical state in people with schizophrenia during various phases of a psychotic episode (Figure 5).

### 4.1. Linking the Level of Thyroid Hormones with the Course of Psychosis

Our results are consistent with current and older studies that showed a reduced T3 level in the group of patients with schizophrenia compared to the group of healthy people [62,63,64] and that T3 levels (total and free) are higher in patients in remission compared to patients without remission [65]. The results of our study, along with others, may suggest that low baseline levels of T3 indicate a poorer response to treatment among patients with psychosis.

This study appears to be the first one to show the effect of suboptimal TSH and thyroid hormone levels on the course of treatment in patients undergoing psychotic decompensation, illustrating the relationship with the number of hospitalizations or the severity of psychotic symptoms at the beginning and during treatment. There is a special role of T3, i.e., a hormone that has a direct impact on the development and metabolism of the brain by regulating the growth of axons and dendrites, myenilization, synaptic formation and the proliferation of specific glial and neuronal populations [66]. Moreover, Mendes-de-Aguiar et al. showed that T3 regulates glutamate uptake and, more specifically, promotes the upregulation of the astrocyte glutamate transporters GLAST and GLT-1, resulting in increased glutamate uptake and thus increased resistance and viability of astrocytes and neurons to glutamate-induced excitotoxicity [67]. Thus, the authors emphasize the role of T3 in improving the astrocyte microenvironment, which is conducive to the development and protection of neurons [67]. T3 may also protect hippocampal neurons from glutamate toxicity through rapid non-genomic regulation [68]. Interestingly, studies also indicate that T3 influences GABA release and reuptake: low T3 concentration increases GABA release induced by depolarization through a direct non-genomic mechanism [69]. Strawn et al. showed that the levels of 5-hydroxyindole acetic acid (5-HIAA) and homovanillic acid (HVA), the main metabolites of serotonin and dopamine, significantly correlated with the level of T3 [70].

In this study, a particular role of the lower TSH values (up to 2.5 mU/L) in comparison to the agreed upper reference ranges (>4 mU/L) emerged. It turned out that the lower the initial TSH level, the greater the severity of positive symptoms such as delusions or hallucinations upon admission to the hospital. At the same time, the lower the TSH values (1–2.5 mU/L), the stronger the relationship with negative and depressive symptoms after a 3 month clinical follow-up. This may suggest that people for whom we have shown a relationship between lower TSH values and negative symptoms after 12 weeks of observation will show a more severe clinical phenotype with the severity of negative and depressive symptoms in early schizophrenia. The probable cause of the observed changes is the positive autoimmunity of the thyroid gland, which, however, cannot be clearly defined due to the lack of assessment of anti-thyroid antibodies among the studied patients. In a recent cross-sectional study, Barbero et al. showed that patients with early psychosis with anti-thyroid antibodies presented the severity of negative symptoms, which was especially visible in the case of anti-thyroid peroxidase antibodies (TPO-Abs), which suggests that thyroid autoimmunity may modulate the severity of negative symptoms [12]. Anti-thyroid antibodies are associated with a more severe phenotype with severe negative symptoms, regardless of thyroid function. Our longitudinal study can complement that mentioned above, as it shows a relationship between the initial lower TSH within the normal range and the severity of negative symptoms that appear several months after the start of treatment.

The study showed a negative relationship between the initial FT3 level and the number of hospitalizations, the intensity of negative symptoms during admission to the hospital and positive symptoms after 3 months of hospitalization, which may indicate a reduced peripheral T4 to T3 conversion among patients with early psychosis [71].

Lower TSH levels in patients with schizophrenia may be associated with disruption of dopaminergic neurotransmission, which may be important as a risk factor for the etiology of this disease [72]. Dopamine inhibits the secretion of TSH in a population of healthy people. Boesgaard et al. suggest that dopaminergic inhibition of basal TSH secretion and TSH pulsation is regulated mainly by D-2 dopaminergic receptors at the pituitary level and, respectively, by D-1 receptors at the hypothalamus level [73]. The level of HVA homovanillic acid, the major metabolite of dopamine, is significantly and negatively correlated with plasma TSH [70]. In a study of patients with exacerbation of schizophrenic symptoms, Rao et al. analyzed the relationship between serum dopamine levels and TSH [74]. It has been found that in patients with schizophrenia, during an exacerbation of symptoms, serum dopamine levels are elevated while TSH levels are reduced. In addition, links between TSH and the serotoninergic system were observed, disturbances of which seem to be important for the course of the disease (see review: [75]). Strawn et al. showed that the variable concentration of 5-hydroxyindole acetic acid (5-HIAA), the main metabolite of serotonin in the cerebrospinal fluid, is significantly and negatively correlated with the level of TSH [70]. 

### 4.2. Relationship of the Level of Thyroid Hormones with the Antipsychotic Drugs (including Partial D2-Agonist Effect) in the Course of Psychosis

Due to the small size of the study group and the diversified pharmacological profile of the used treatment, the division into subsequent groups depending on the different classes of antipsychotic drugs was statistically impossible, nevertheless at this stage of the research, it cannot be ruled out that the obtained results were influenced by the pharmacotherapy. However, the results of individual analyses on this subject are contradictory and inconclusive [76].

There are limitations to the use of equivalent doses of second-generation drugs for different antipsychotics. The available literature on the subject indicates the possibility of a different effect of individual antipsychotic drugs on endothelial function and inflammation. Nevertheless, analyses based on the variable pharmacological profile of the effects of single drugs will be part of another work on a larger group of patients.

However, when referring to individual cases of patients taking certain antipsychotic drugs, it can be noted that in the case of aripiprazole (the most commonly used partial D2 receptor agonist), its influence on TSH levels can be treated quite differently. There are data that show that, for example, chlorpromazine (a typical neuroleptic D2 receptor antagonist) causes a significant decrease in TSH levels after discontinuation. The use of aripiprazole (D2 partial agonist and 5-HT2A antagonist) may therefore have a similar effect to chlorpromazine withdrawal, i.e., it may reduce TSH levels [77]. On the other hand, other studies show that there are no significant changes in TSH and FT4 levels for aripiprazole.

When aripiprazole is combined with quetiapine (showing, among others, a significant ability to bind to D2 receptors), it is known that since quetiapine itself can induce symptomatic hypothyroidism (which is mainly caused by hypothalamic/pituitary dysfunction in central [secondary] hypothyroidism) [78], their combination may give an increased effect. It should be emphasized, however, that in the case of quetiapine the biological half-life is approx. 7 h, and in the case of aripiprazole, it is 3–5 h, while patients were qualified for the TSH blood test 8 h after the last dose of the drug.

Current studies support the conclusion that the use of atypical antipsychotics, such as clozapine, olanzapine and quetiapine, is not associated with a significant increase in pituitary prolactin. This explains the dopamine-sparing mechanism in the brain’s neoplastic funnel, the dopamine pathway that controls prolactin secretion. However, the factors increasing the secretion of prolactin include, among others, TSH, mental or physical stress. Current data indicate that both conventional antipsychotics and high doses of risperidone (>6 mg/day) increase prolactin levels to the apparent clinical symptoms of hyperprolactinemia. Nevertheless, it is believed that the lack of prolactin elevation reported with atypical antipsychotics is due to their much greater specificity, which results in less blockade of dopamine receptors in the tuberoinfundibular pathway [79]. For this reason, treatment with aripiprazole is in most cases associated with decreased prolactin levels [80].

In contrast, monitoring of thyroid function is recommended mainly in patients taking antipsychotic drugs who, contrary to the abovementioned theories, show high levels of prolactin and who are at risk of thyroid abnormalities [77,81], which was also included in the exclusion criteria of our study.

There are isolated cases combining aripiprazole with haloperidol, based on which the authors indicate a probable increase in the content of pituitary GH and thus an increased release of TSH, explaining it by the dopaminergic blockade of haloperidol, causing an increase in TSH in the pituitary gland and, subsequently, in the blood [82]. However, based on the evaluation of [3H]spiperone binding assays to D2 receptors and prolactin mRNA on the pituitary glands of rats that were daily injected orally with either haloperidol (2 mg/kg) or aripiprazole (24 mg/kg) for 21 days, there was a noticeable increase in the level of receptor expression D2 by 41% and 38% after haloperidol, with a simultaneous increase in prolactin by 26%. In contrast, concomitant treatment with aripiprazole-reduced [3H]spiperone binding by 24%, while D2L and D2S mRNA levels decreased by 23% in both cases, showing no effect on pituitary prolactin mRNA [83].

The study comparing the use of clozapine and haloperidol and their effects on thyroid hormones found that the clozapine-treated group showed a blunted TSH response, while patients treated with haloperidol had a normal TSH response to TRH [84]. 

In the case of treatment with olanzapine, which is often associated with weight gain and metabolic disorders, its unfavorable metabolic effect was abolished in people with a low TSH profile, who were most sensitive to the adjuvant treatment of psychosis, blocking olanzapine-induced weight gain [85]. However, these data are not reflected in our two patients, who in the first week of aripiprazole monotherapy after starting olanzapine at 12 weeks of observation with a lower baseline TSH level (<2.2) showed an exaggerated metabolic effect associated with weight gain and an increased lipid profile (↑ BMI, ↑ TC) after 12 weeks of observation. The positive symptoms of psychosis were nevertheless suppressed.

These observations may be explained by the variable individual profile dependent on polymorphisms in responses to olanzapine and risperidone. Olanzapine is mainly metabolized by the enzymes of cytochrome P450, CYP1A2 and CYP2D6, while the metabolism of risperidone is mainly mediated by CYP2D6 and CYP3A4. Polymorphisms in these genes, other enzymes and their transporters show differences in pharmacokinetics [86]. In addition, the available literature indicates the possibility of a different effect of individual antipsychotics on endothelial function and inflammation.

The preferred method of comparing drug doses is through logistic regression to compare each drug to chlorpromazine or haloperidol to create a formula for each existing drug relationship. It is worth noting that these methods would only make any sense if the group size were a minimum of 33 (necessary to maintain test power and required to perform logistic regression).

Based on ‘best dose’ measures, to compare the effectiveness of a drug under different conditions, treatment should be based on empirical quantitative models that can be used to equalize the doses of the drugs. However, the proposed measure has many shortcomings. Using this method, it was found, for example, that haloperidol has a ratio of 1.6 (corresponding to 100 mg of chlorpromazine), which, however, cannot be easily transferred to our study because patients diagnosed with schizophrenia showed a different profile of drug metabolism (related to the inherent profile of CYP2D6 and CYP1A2) and medication intake at different times.

Our observations related to symptomatic improvement are also supported by the studies by Nazou et al., in which, after the administration of olanzapine at a dose of 5 mg/day in psychotic patients, there was a significant reduction in the levels of thyroid hormones with the induction of Hashimoto’s thyroiditis. After discontinuation of olanzapine and its replacement with 75 μg/day levothyroxine, psychotic symptoms improved significantly [87].

There is also speculation that the combination of olanzapine and aripiprazole may lead to a greater partial agonist effect than its antagonistic effect at the D2 receptor, resulting in an exacerbation of psychotic symptoms [77].

During the first week of treatment, 17 patients of our study were taking haloperidol, including one with the addition of aripiprazole. Eight patients were taking olanzapine. One patient took risperidone and one quetiapine with aripiprazole. One patient was taking aripiprazole as monotherapy. Eleven patients were initially off medication. At the second time point, 14 patients were taking two neuroleptics; 8 of them—quetiapine, 12—aripiprazole and 4—olanzapine. In 14 patients, monotherapy was conducted with olanzapine, in 5 with quetiapine, in 3 with risperidone; one patient was treated with haloperidol, and one with clozapine.

As a result, especially in the second time point, we dealt with a dozen or so different combinations of a first-generation and second-generation drug, which makes it extremely difficult to conduct analyses other than those relating to the conversion of pharmacotherapy into chlorpromazine equivalents.

It should be noted that neuroleptics differ in their effects on thyroid function and TSH levels. However, it is difficult to reach a final conclusion in this respect because the literature on the subject does not include extensive meta-analyses, but single studies, case studies, or studies on animal models predominate. It may also be important to determine whether this influence appears in the case of people with normal or already disturbed thyroid function. Psychosis itself, together with comorbid diseases, will have its own importance in this subject. There is no data on the synergistic or cumulative effect of individual drugs or the cancellation of opposite effects. The impact may also be different at different stages of the treatment. 

### 4.3. Relationship of the Level of Thyroid Hormones with Depressive Symptoms in the Course of Psychosis

It has been shown that lower TSH levels are significantly associated with greater severity of depressive symptoms after 3 months of hospitalization in patients with psychotic decompensation on admission to the hospital. It suggests that lower TSH values may be a promising biomarker of the risk of severe depressive symptoms in patients with psychosis even after months after starting antipsychotic treatment. To our knowledge, the presented study is the first in which we show the influence of lower TSH values on the severity of depressive symptoms in the course of psychotic decompensation. In the literature on the subject, one can find a study conducted in patients with an acute psychotic episode in the course of schizophrenia, where no relationship was found between the level of TSH and the severity of symptoms (including negative or depressive symptoms) during admission to hospital or 6 weeks later [88]. In a systematic review of depressive symptoms and suicidal behavior in the period after the first psychosis, many factors increasing the risk of depressive symptoms were distinguished, but so far, no association of decreased TSH levels during hospitalization were identified [89].

This is confirmed by large studies on a group of healthy young people (n = 6869), where current depressive symptoms were associated with subliminal lower TSH levels [90]. Similarly, large population studies of 12,787 healthy subjects confirmed that people with lowered TSH more often suffer from subclinical depression [91]. Low TSH levels in healthy people with a median age of 23 years are associated with an increased risk of developing depression [92]. Our results are consistent with the studies by Medici et al., who proved that people with low-normal levels of TSH have a higher level of comorbid depressive symptoms, as well as a significantly increased risk of developing a depressive syndrome in the following years [93].

Although few literature reports indicate that the previously present low TSH level may induce depressive symptoms, one study found a relationship between low baseline TSH levels and the severity of depressive symptoms after 6 months of observation in the elderly group [94]. Similarly, in another study, patients with low normal TSH levels but no depressive symptoms at baseline were at increased risk of developing a depressive syndrome in the following years of the 8 year study [93]. Moreover, cross-sectional studies conclude that depression is more common in patients with low-level serum TSH. Depressive states are more common in women with low TSH hospitalized in psychiatric hospitals [95]. Although thyroid dysfunction was shown to influence the risk of depression, depression itself can also affect thyroid parameters [96,97]. It should be noted that in patients with mood disorders, thyroid hormones modulate the expression of phenotypic depression during the observation period. It can therefore be concluded that a low normal level of TSH is an important risk factor for the development of depression in the shorter and longer term. Due to the fact that prospective studies are scarce and there is a lack of studies among patients with schizophrenia in this area, it seems important to study the relationship between thyroid function and depression not only in a cross-sectional manner but also prospectively.

The most reproducible abnormality in patients with depression is a weaker response to morning TRH in terms of TSH levels and a disturbance in the diurnal TSH secretion consisting in the lack of a nocturnal increase in TSH levels [98], which may explain the reduction in TSH levels in patients with symptoms of depression. The relationship of the affective component with deviations in the level of TSH is supported by a study by the same author, which showed a different response to TSH secretion in the protirelin (TRH) test at night and in the morning in patients with schizoaffective disorders in the depressive phase, but not in patients with schizophrenia [99].

Importantly, our results indicate that even small changes in the functioning of the thyroid gland within the normal range can have a significant impact on both negative and affective symptoms in patients undergoing psychotic decompensation. The presented data suggest that although the assessment of TSH level and thyroid functioning on the surface seems to be correct and does not differ from the statutory reference values, the adopted cut-off values in established clusters (1, 2 for lower and higher TSH values < and >2.5 mU/L) mark the limits of the clinical horizon related to the severity of depressive symptoms in the group of people with psychosis.

### 4.4. The Relationship of Thyroid Hormones with the Intensification of the Inflammatory Process

A significant correlation has been shown between the reduced level of TSH within the normal range at the time of admission to the hospital and negative symptoms such as social isolation or blunting of effect in patients after 3 months of treatment. The relationship shown by us is important because it can help to understand the mechanisms underlying the development of negative symptoms, and due to the fact that standard antipsychotics have limited therapeutic benefits for this group of symptoms, therefore there is a significant need to develop new treatment options targeting negative symptoms [100]. The mechanisms of reduced TSH levels in patients with aggravated negative symptoms are most likely multiple, and pro-inflammatory cytokines or anti-thyroid antibodies may play an important role. The neuro-inflammatory theory [101] emphasizes the relationship between pro-inflammatory cytokines and the intensification of negative symptoms in schizophrenia [102,103,104,105]. This is evidenced by a strong positive relationship between the initial levels of tumor necrosis factor-α (TNF-α) and interleukin 6 (IL-6) with the development of negative symptoms [104]. At the same time, it was found that TNF-α induces a significant decrease in TSH levels [106]. Similarly, interferons (IFNs) induce profound changes in thyroid hormone metabolism, including a significant decrease in TSH, possibly mediated in part by IL-6 [107]. The literature on the subject also draws attention to the possible role of interleukins: 1β, 7, 8, matrix metalloproteinases (MMP), cortisol in the cerebrospinal fluid, blood serum and salivary albumin and cortisol as indicators of inflammation related to the course of the disease in people with schizophrenia [57,108]. There are few studies comparing the levels of the above-mentioned biomarkers during the development of the disease, explaining the molecular mechanism of psychosis [109]. In the metabolomic studies of the saliva of patients with the first episode of psychosis, attention was drawn to a reduced metabolism of aromatic amino acids, altered glutamine metabolism and increased metabolism of the tricarboxylic acid cycle, compared to those with a high clinical risk of psychosis before its occurrence [110,111,112,113,114]. It showed a profile of changes with greater intensity at the onset of schizophrenia, characteristic of the disease stage. A variety of salivary metabolites can be regarded as potential diagnostic biomarkers to indicate the severity of the disease. An additional argument supporting the use of salivary markers in the diagnosis of psychosis is their relationship with the peripheral indicators of the inflammatory response, oxidative stress, damage to the blood–brain barrier and salivary microflora [43,44,108,115,116]. Currently conducted research on the oral cavity microbiome draws attention to its close relationship with the brain, which indicates the contact between salivary metabolites and the brain. The salivary microbiome of patients with the onset of schizophrenia is characterized by heterogeneity and might be dependent on the duration of the disease. Oral metabolism disorders often precede the actual onset of schizophrenia, increase with the development of the disease and even could be its causative factor. Cui et al. stressed that certain salivary microbiota exhibited disease-specific correlation patterns with symptomatic severities [117]. It might be explained by the production of pro-inflammatory cytokines and local immunological imbalance, amino acid metabolism, carbohydrate metabolism and xenobiotic degradation. Qing et al. gave an explanation of how damaged oral microbiota might reach the brain through abnormal metabolites, i.e., via the olfactory tract and through the blood–brain barrier [118]. It may lead to local redox imbalance and cause the initiation of schizophrenia.

The correlation between salivary and peripheral inflammatory markers or redox systems indicates the potential importance of the oral-brain connection in the pathogenesis of schizophrenia. Due to the multitude of direct antioxidants in saliva, the measurement of the concentrations of specific substances would be time-consuming and costly, therefore methods to measure the net antioxidant effect of all direct antioxidants (TAC expressed as FRAP) are used more often. Salivary TAC is influenced by a whole range of endogenous substances such as glutathione, uric acid, vitamins, some proteins and exogenous dietary substances such as polyphenols, drugs, microelements such as selenium, etc. In the case of selenium, it is necessary for the synthesis of triiodothyronine (FT3) and, therefore, must be supplied in the daily diet. In addition to supporting the proper functioning of the thyroid gland, selenium is also a key component of several functional selenoproteins, the most famous of which are antioxidant enzymes, e.g., glutathione peroxidase (GPx), whose main role is to remove hydrogen peroxide, harmful lipid hydroperoxides produced in vivo by organisms of aerobic origin. GPx deficiency exacerbates endothelial dysfunction, the main factor associated with the unsealing of the blood–brain barrier and the intensification of symptoms of cardiovascular failure also observed in the course of psychotic decompensation. The explanation for the observed positive relationship between salivary FRAP and FT3 is the salivary compensation mechanism in relation to the neurotoxic effects of elevated glutamate levels, which impairs the functions of the N-methyl-d-aspartate (NMDA) receptor, which leads to the appearance of negative symptoms and cognitive dysfunctions in patients with schizophrenia. The visible negative correlation between FRAP, FT3 and negative symptoms is an expression of the aforementioned compensation of salivary antioxidant systems in relation to the glutamate-induced neuronal toxicity.

### 4.5. Relationship of Thyroid Hormones with the Results of Neuroimaging Studies

In our work, we showed a negative relationship between TSH, FA right frontal lobe DEV (diffusion in the right frontal lobe) and ACC_MI (myoinositol in the anterior cingulate cortex). The changes were seen in the voxel-based, and diffusion tensor-based morphometry imaging study suggest a negative association of morphological and microstructural brain lesions with TSH levels in early psychosis (in a voxel-diffusion-tensor-based morphometry imaging study; such dependence may be related to various factors that damage CSN, e.g., alcohol, psychoactive substances, despite that such patients were excluded from the study). However, there are no studies explaining the observed relationship between the functioning of the hypothalamic–pituitary–thyroid axis with microstructural abnormalities in psychotic patients. The negative correlation between TSH and DEV may be related to the limited movement of micronutrient particles from astrocytes to neurons within the right frontal lobe. In addition, we can assume that in this area, there is also damage to the white matter and reduction of connections between neurons [119], which would confirm the hypofrontality thesis in schizophrenia [120]. In accordance with the literature on the subject, disturbances in the connections of neural networks in the frontal lobes are responsible for some clinical symptoms, including emotional control, motivation to act and the occurrence of negative symptoms [121]. 

This is also supported by the second negative correlation between TSH and the level of myoinositol in the ACC, which could be explained by brain tissue edema of examined areas (although the measurements of thickness or gray to white matter ratios were not performed), which could be induced by a decreased flow of components in analyzed brain regions. This situation resembles a cascade of events during an ischemic stroke. DWI has found particular application in the diagnosis of early and very early ischemic changes in the brain. In improperly supplied brain cells, for example, as a result of an early stroke or transient ischemia, the permeability of cell membranes changes, which most likely also occurs in the presented research in the group of analyzed patients in early psychosis. The dependencies between TSH and myoinositol in ACC shown by us, according to the literature data, indicate astroglial dysfunction in this brain area, which is mainly responsible for the pathogenesis of the disease according to one of the theories [122]. According to the literature on the subject, damage to astrocytic cells leads to a disturbance of the redox balance, a reduction in the functioning of the immune system by the use of glial cells [123], a reduction in the secretion of synaptogenic factors involved in the formation of synapses [124] and a reduction in synthesis L-serine, which is a substrate for D-serine building NMDA receptors [125]. Thus, damage to astrocytes also leads to glutamatergic dysregulation, which was proved crucial in endophenotypes of schizophrenia [30]. The ACC region is extremely important due to the fact that its abnormalities constitute the neurobiological basis of many clinical symptoms of schizophrenia, which is confirmed by neuropathological and neuroimaging results [126].

Scientific data emphasize that the increase in myoinositol in the CNS occurs in the case of TBI (traumatic brain injury) [127] and microglial cell damage resembling the cascade occurring in Alzheimer’s disease [128]. We have observed the correlation between FT4 and MI/CR. Myoinositol is a marker of glial cells representing the membrane transport of nutrients for neurons that previously was described in the therapy of autoimmune thyroiditis [129]. MI can decrease the level of TSH and FT4, which is poorly described in the context of glial development and physiology. However, thyroxin participates in the mediation of signals in astrocytes [130] and is essential for cells motility, especially glial cells and neurons [131]. MI reflects glial activation in patients with psychosis. Revealed correlation might be a new goal in further studies and also a clue point between thyroxine action on the glial cell through myoinositol, which is also very important in the pathogenesis of schizophrenia, as emphasized by scientific data [132,133]. Activation of microglial cells thus contributes to the disturbance of the immune balance and intensification of inflammation in the CNS through the production of IL-1, IL-6 or TNF-α, which was confirmed in this group of patients [134].

Both the assessment of the thyroid and myoinositol by MRS can be a useful tool for stratifying and examining such patients. The observed relationship of microglia with FT4 is supported by the observed positive correlation with FT3. Glutaminergic dysregulation, visible in our previous studies, which turned out to be key in schizophrenia endophenotypes [30], is supported by thyroid function and its positive relationship with FT3 production, which confirms the demonstrated relationship between FT3 and GLU/CR in ACC.

The relationship between the antioxidant system of saliva (positive relationship with FRAP) and FT3 concentration, observed in our research, supports the neuro-inflammatory theory of early schizophrenia [135], activation of lymphocytes (positive relationship with TSH) and neutrophils (positive relationship with FT4).

### 4.6. The Importance of Early Childhood Trauma

In the literature on the subject, there are no studies on the relationship between the functioning of the thyroid gland and the history of childhood trauma in the population of people with schizophrenia, despite the fact that the experience of trauma in this group is a frequent phenomenon, i.e., patients with psychosis report adverse experiences in childhood almost three times more often than in control subjects [20]. The history of childhood trauma affects the clinical picture and the course of treatment in people with schizophrenia [21,22,23,24,25,26]. To our knowledge, the presented study is the first to illustrate the functioning of the thyroid gland depending on the psychosocial factor, which is the experience of childhood trauma in a group of people with early psychosis. We found a positive correlation between the experience of both emotional abuse and physical neglect in childhood and the value of the baseline FT4 level, which suggests that the history of childhood trauma leads to an increased role of the HPT axis in the pathomechanism of the observed changes. Early in life seems to be a particularly sensitive time to long-term changes in the endocrine system. Animal studies have shown that weaker maternal care associated with insufficient tactile stimulation results in higher T4 in the offspring [27]. Much of what we know about thyroid hormone changes following human trauma comes from research on post-traumatic stress disorder (PTSD). Increased levels of total and free T3 and an increase in total T4 levels were observed among PTSD war veterans compared to the control group [136,137,138]. In recent population studies where biochemical measurements were taken one year before and one year after an earthquake, an increase in free T4 concentration was found [139]. Hormonal studies of people with a history of childhood trauma are rare. Associations between thyroid hormones (free and total T3) and the severity of PTSD symptoms were reported in girls who have experienced sexual abuse [140]. In the studies by Machado et al. among adolescents, it was shown that a decreased level of T3 is associated with a history of trauma early in life [28]. Although research on the relationships of the thyroid gland and the experience of trauma is sparse and the results are sometimes inconsistent, a general conclusion can be drawn that the experience of childhood trauma is characterized by abnormalities in the functioning of the hypothalamus, pituitary and thyroid axis and altered levels of thyroid hormones. In the early psychosis population, elevated fT4 levels may represent the long-term thyroid consequence of childhood trauma.

### 4.7. Limitations

This work has a number of limitations. Note the small sample size due to the strict inclusion/exclusion criteria. A larger group would allow taking into account the sex and age factors for the examined variables differentiating from the HPT axis. 

Another significant limitation of this study is a statistically significant link, but with a low number of patients; hence the need to conduct studies on a larger study group in order to be able to try to predict the clinical course of the disease to a greater extent. Future studies would verify whether the results obtained in this study are correct and how accurate TSH is in predicting the disease.

Another limitation is the difficult assessment of the influence of the treatment on the level of TSH due to the small size of the studied group. Division into subsequent groups would not allow such analyzes to be carried out. As a result, especially in the second time point, we dealt with a dozen or so different combinations of a first-generation and second-generation drug, which makes it extremely difficult in the study group to conduct analyses other than those relating to the conversion of pharmacotherapy into chlorpromazine equivalents.

It seems interesting to determine in our future studies whether, for example, on the basis of TSH levels, it is possible to predict the therapeutic effect of the drugs used or the quality of life of the studied patients.

Another limitation is the lack of a detailed assessment of thyroid function (assessment of the concentration of anti-thyroid antibodies to thyroid peroxidase: anti-TPO and/or anti-thyroglobulin, anti-Tg). However, according to studies by Sæther et al., the level of anti-TPO and anti-Tg in patients with non-affective psychosis and recurrent depressive episodes did not differ from the control group, while in the group of non-affective psychoses, the differences related to it were found only in approx. 6% and 1% of respondents, respectively [141]. Hence, the hypothetical lack of the possibility of a relationship between these antibodies and the indicators tested by us seems equally probable. None of the patients presented signs of thyroid pathology in the physical examination. However, a limitation of the conducted tests is the lack of additional analyses of a general assessment of thyroid function, such as imaging tests (ultrasound of the thyroid gland).

Verification of thyroid hormones in patients with schizophrenia could be a problem. It is not easy to check thyroid hormones levels in patients changed by the psychoactive process. It might be explained by the variety of clinical symptoms, heterogeneity of patients, smoking, unhealthy lifestyle, taking drugs or psychoactive substances, co-existence of cardiovascular pathology, etc. All mentioned states could distort the level of hormones, causing its fluctuation. Patients with schizophrenia that have thyroid dysfunction in the acute period as well as at the chronic stage are not rare. It does not always depend on co-existing pathologies. It was previously described that thyroid hormones impact the network of dopaminergic, serotonergic, GABAergic and glutaminergic neurotransmitters [72,75,142,143,144]. These interactions in patients with schizophrenia concern its etiological aspects. The abnormal regulation of the mentioned networks correlates with psychiatric diseases. We attempted to clarify the possible influence of the thyroid gland on schizophrenia. We showed that thyroid hormones can serve as an additional biomarker in schizophrenia as well as to improve diagnostic or therapeutic procedures.

The study included a group of patients in the age range from 15 to 45 who require acute psychotic decompensation requiring hospitalization in the course of schizophrenia. The results obtained in different age groups or in patients in remission may be different.

The diagnosis of schizophrenia, despite the progress in scientific knowledge, is still based on the unreliable criterion of the clinical picture. Hence, one should assume the heterogeneity of the pathophysiology, pathomorphology and etiology of symptoms in the study group. The group covered by the study may not be sufficient to select all possible subgroups within the studied range.

Further research on a larger sample of psychotic patients with various clinical phenotypes may constitute a premise for an in-depth characterization of the molecular mechanisms associated with disturbances in the functioning of the HPT axis and their relationship with clinical, imaging or laboratory variables, facilitating the search for objective biomarkers of the course and prognosis of the progress of schizophrenia as well as more effective forms of its therapy.

## 5. Conclusions

In the presented study, we noted that the lower TSH cut-off levels, with an upper limit of 2–2.5 mU/L, were the most statistically significant in relation to severe depressive symptoms after three months of early psychosis therapy (BDI_12; ROC, AUC = 0.8). The relationships between TSH and brain imaging based on the diffusion tensor in the right frontal lobe FA right frontal lobe DEV and the level of myoinositol in the ACC_MI cingulate gyrus (negative association of morphological and microstructural changes in the brain with the level of TSH) proved to be equally important. Although there are no studies explaining the observed relationship between TSH and brain microstructural abnormalities in patients with psychosis, the results of this study indicate that the limited movement of micronutrient particles from astrocytes to neurons within the right frontal lobe may be a pathogenetic factor of white matter damage and reduced connections between neurons. This would support the hypofrontality theory of schizophrenia, corresponding to behavioral and cognitive deficits dependent on spontaneous default activity associated with frontal lobe dysfunction. According to the literature on the subject, disturbances in the connections of neural networks in the frontal lobes are responsible for some of the clinical symptoms, including negative symptoms or depressive symptoms. Previous attempts to study the function of the frontal lobe in schizophrenics have focused on neuropsychological tests, EEG, cerebral blood flow (CBF) scans or positron emission tomography (PET), indicating abnormalities between the schizophrenic group and the control group. However, the relationships between the pituitary, the released TSH and the brain imaging based on the diffusion tensor in the FA right frontal lobe DEV demonstrated in this study are the first that objectively explain the mechanism of the observed changes in early schizophrenia (in the clustering method without prior available knowledge).

The results of this study also indicate a strong positive relationship between FT3 and salivary total antioxidant potential (FRAP), which presents a compensatory mechanism of salivary antioxidants to glutamate-induced neuronal toxicity correlated with severe depressive symptoms on admission to the hospital. Nevertheless, the oxidative-antioxidant system related to the HPT axis may also play an inverse role related to the involvement of PON-1 in the metabolic side effects associated with treatment with second-generation neuroleptics (SGA). In the presented study, a positive correlation was found between PON-1 activity, HDL levels and the concentration of FT3 after 12 weeks of pharmacotherapy. This may prove that schizophrenia is a neuro-immune state related to the toxicity of oxidative stress, which is a part of the pathophysiology of this disease.

In the presented work, we redefine the commonly accepted cut-off ranges for TSH in relation to the obtained clinical phenotypes of patients in early schizophrenia. These data provide evidence suggesting the need for early therapeutic intervention for lower TSH levels, for which the mental state of people in early psychosis may herald clinical improvement or suggest the need to modify the psychotic treatment plan in order to mitigate the adverse effects associated with increased depressive symptoms. This is in line with the current majority of international guidelines indicating the need for an individualized treatment plan adjusted to the patient’s age, but also the degree of change in serum thyroid-stimulating hormone (TSH) levels, reported symptoms, the risk of cardiovascular diseases and/or other comorbidities [15,145].

Apart from the cognitive values related to the influence of subtle differences in TSH levels on the patient’s clinical picture in early psychosis, an innovative presentation of the above-mentioned relationships in the context of the experience of childhood trauma is an additional advantage of this work. In connection with the obtained research results, it participates in the presentation of mechanisms dependent on the functioning of the HPT axis.

## Figures and Tables

**Figure 1 jpm-12-00247-f001:**
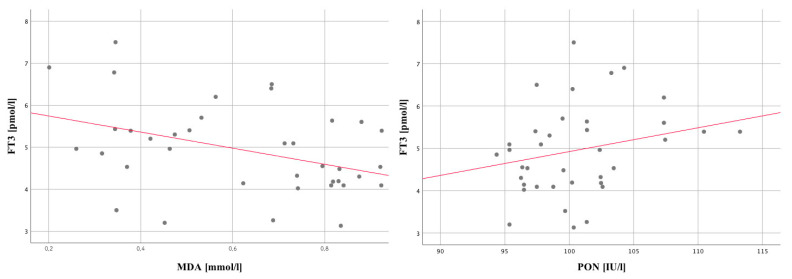
Relationship between FT3 and MDA/PON, respectively.

**Figure 2 jpm-12-00247-f002:**
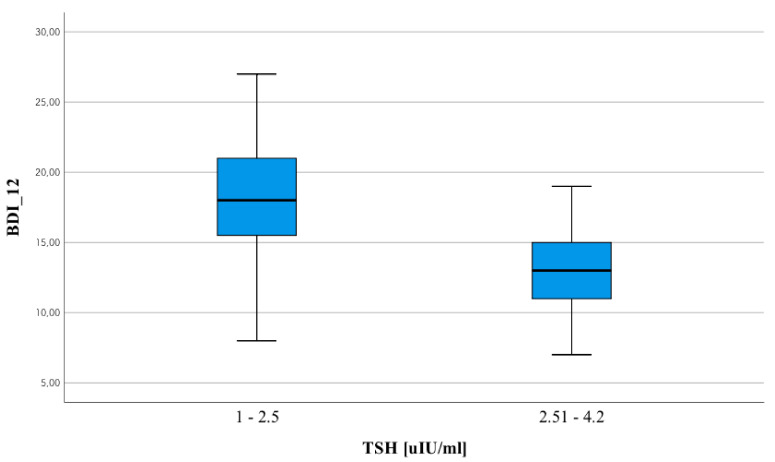
BDI_12 scoring in groups of people divided according to the level of TSH.

**Figure 3 jpm-12-00247-f003:**
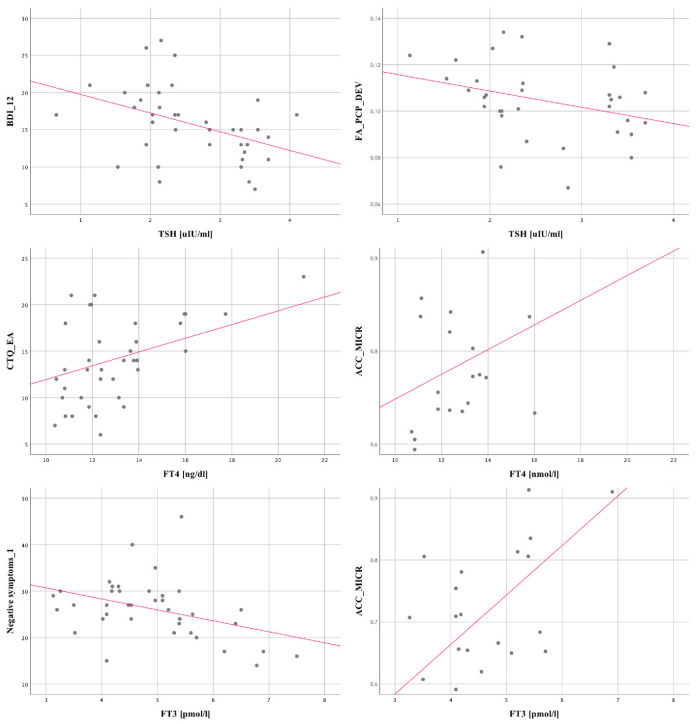
Relationship between the functioning of the thyroid gland and individual variables.

**Figure 4 jpm-12-00247-f004:**
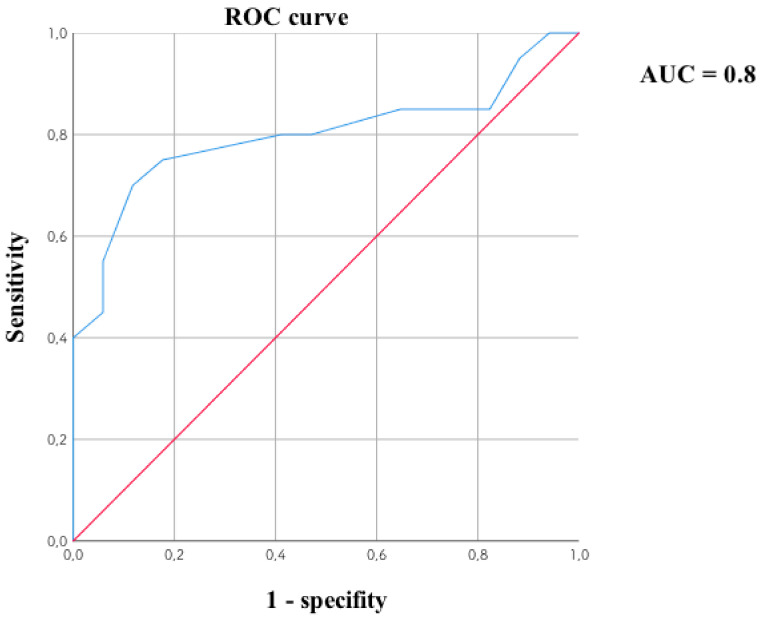
ROC curve for BDI_12. The TSH level was tested once, and the BDI was measured after 12 weeks. The ROC curve is related to the BDI_12 prediction based on the TSH level.

**Figure 5 jpm-12-00247-f005:**
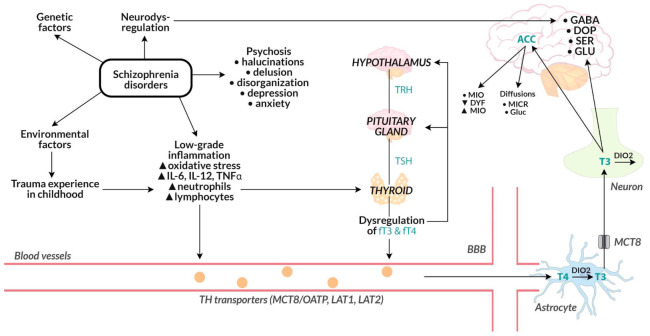
Relationship between schizophrenic disorders and HPT axis. DIO2—iodothyronine deiodinase 2; DOP—dopamine; DYF—diffusion; FT3—free triiodothyronine; FT4—free thyroxin; GABA –γ-aminobutyric acid; GLU—glutamate; Gluc—glucose; IL—interleukin; LAT—L-type/large neutral amino acid transporter; MCT8—monocarboxylate transporter 8; MIO—myoinositol; MICR—myoinositol creatine; OATP—organo-anion-transporter; SER—serotonin; T3—triiodothyronine; T4—thyroxin; TH—thyroid hormones; TNFα—tumor necrosis factor α; TRH—thyrotropin-releasing hormone; TSH—thyrotropin.

**Table 1 jpm-12-00247-t001:** Research methods used in the study.

Method	Reference	Description
**Clinical data**		
Clinical history	n/a	Duration of untreated psychosis (DUP), number of episodes (QE), age of onset (AE), course of the first episode (AF), number of hospitalizations (QH), hospitalization length (HL).
PANSS (Positive and Negative Syndrome Scale)	[35]	A tool for assessing the severity of schizophrenia symptoms. Thirty items rated on a scale from 1 (asymptomatic) to 7 (extremely symptomatic) considering five factors: positive symptoms (PANSS pos), negative symptoms (PANSS neg), disorganized thoughts (PANSS dis), uncontrolled hostility/excitement (PANSS exc), anxiety/depression (PANSS emo).
BDI-II (Beck Depression Inventory, second ed.)	[36]	The scale of severity of depressive symptoms used in psychiatrically diagnosed patients. Includes 21 factors scored from 0 to 63. Cut-off points: minimal depression (0–13), mild depression (14–19), moderate depression (20–28), severe depression (29–63).
CDSS (Calgary Depression Scale for Schizophrenia)	[37,38]	Tool for assessing depressive symptoms in people with schizophrenia, including 9 issues assessed from 0 to 3 (total: 0–27). An increase in the score means an increase in the severity of depressive symptoms. A score above 6 has an 82% specificity and 85% sensitivity for predicting the presence of a major depressive episode.
CTQ (Childhood Trauma Questionnaire)	[39,40]	Specifies information about the experience of trauma in childhood and adolescence. Five trauma types subscales: emotional abuse (CTQ-EA), physical abuse (CTQ-PA), sexual abuse (CTQ-SA), emotional neglect (CTQ-EN), physical neglect (CTQ-PN), and the total trauma (CTQ-Total) scale, which is the sum of the individual subscales. An increase in the score indicates an increased risk and severity of the experience of trauma.
**Routine laboratory tests including markers of HPT axis function**		
Routine laboratory tests	n/a	Thyroid and HPT markers (TSH, FT3, FT4), inflammation markers (CRP, ESR), ionogram (K^+^, Na^+^, Mg^2+^), full blood count, metabolic parameters (glucose and lipid profile: TC, HDL, LDL, TG), renal function markers (creatinine, urea, eGFR), liver function markers (AST, ALT, GGTP).
**Specialized tests (performed outside the standard test protocol)**		
Specialized laboratory tests	[41,42,43,44]	The total antioxidant potential expressed as FRAP in saliva and blood serum, serum malondialdehyde (MDA), serum paraoxonase 1 (PON-1).
Specialized neuroimaging studies	n/a	MRI of the brain (T1W, T2W, FLAIR, DWI, SWI sequences), 1H-MRS for the quantitative and qualitative evaluation of the spectrum of neurometabolites (i.e., myoinositol).
Pharmacotherapy	n/a	Drugs used converted into a chlorpromazine equivalent

AST: aspartate aminotransferase; ALT: alanine aminotransferase; CRP: C-reactive protein; eGFR: estimated glomerular filtration rate; ESR: erythrocyte sedimentation rate; FT3: free triiodothyronine; FT4: free thyroxine; HDL: high-density lipoprotein; GGTP: γ-glutamyltranspeptidase; LDL: low-density lipoprotein; TC: total cholesterol, TG: triglycerides; TSH: thyrotropin.

**Table 2 jpm-12-00247-t002:** Relationship of thyroid parameters with the course of the disease, clinical condition at week 1 and 12 and childhood trauma.

Variable	TSH		FT4		FT3	
	r_s_	*p*	r_s_	*p*	r_s_	*p*
DUP	0.17	0.29	0.29	0.08	−0.12	0.47
Number of hospitalizations (QH)	−0.07	0.68	0.17	0.30	−0.36 *	0.03
Age of onset (AE)	0.09	0.59	−0.25	0.13	0.21	0.21
Number of episodes (QE)	0	0.98	0.05	0.76	0	0.98
Hospitalization length (HL)	−0.01	0.96	0.14	0.40	−0.07	0.69
PANSS pos_1	−0.33 *	0.04	0.02	0.91	−0.14	0.39
PANSS neg_1	0.05	0.74	−0.19	0.25	−0.39 *	0.01
PANSS dis_1	−0.22	0.18	0.28	0.09	0.01	0.97
PANSS exc_1	0.08	0.64	0.01	0.96	−0.25	0.14
PANSS emo_1	0.17	0.31	−0.22	0.19	0.17	0.31
PANSS pos_12	0.01	0.95	0.09	0.58	−0.36 *	0.03
PANSS neg_12	−0.18	0.28	−0.12	0.49	−0.12	0.50
PANSS dis_12	−0.20	0.22	0.15	0.38	−0.17	0.32
PANSS exc_12	0.20	0.23	−0.15	0.39	−0.02	0.89
PANSS emo_12	0.03	0.84	−0.06	0.73	0.11	0.51
BDI_1	−0.18	0.29	−0.07	0.66	0.23	0.16
BDI_12	−0.44 **	0.01	0.05	0.77	0.18	0.29
Calgary_1	−0.19	0.24	−0.09	0.61	−0.05	0.78
Calgary_12	−0.28	0.09	0.05	0.75	−0.16	0.33
CTQ_EN	−0.19	0.24	0.30	0.07	−0.09	0.59
CTQ_EA	−0.14	0.41	0.46 **	0	−0.11	0.52
CTQ_PN	−0.26	0.12	0.39 *	0.02	−0.01	0.95
CTQ_PA	−0.13	0.43	0.31	0.06	0.01	0.97
CTQ_SA	−0.26	0.11	0.28	0.08	0.09	0.60
CTQ_TOTAL	−0.17	0.31	0.38	0.02	−0.10	0.55

*—*p* ≤ 0.05; **—*p* ≤ 0.01;_1—results during the first week of hospitalization; _12—results three months after admission to the hospital; BDI—Beck Depression Inventory; Calgary—Calgary Depression Scale for Schizophrenia; CTQ—Childhood Trauma Questionnaire; CTQ_EA—emotional abuse; CTQ_EN—emotional neglect; CTQ_PA—physical abuse; CTQ_PN—physical neglect; CTQ_SA—sexual abuse; CTQ_TOTAL—total trauma; FT3—free triiodothyronine; FT4—free thyroxin; *p*—*p*-value; PANSS—Positive and Negative Syndrome Scale; PANSS pos—positive symptoms; PANSS neg—negative symptoms; PANSS dis—disorganized thoughts; PANSS exc—uncontrolled hostility/excitement; PANSS emo—anxiety/depression; r_s_—Spearman correlation; TSH—thyrotropin.

**Table 3 jpm-12-00247-t003:** Relationship of thyroid function assessment markers with routine parameters at 1 and 12 weeks of clinical follow-up.

Variable	TSH		FT4		FT3	
	r_s_	*p*	r_s_	*p*	r_s_	*p*
WBC (×10^3^/µL)_1	−0.11	0.50	0.08	0.64	0.17	0.30
NEUT (×10^3^/µL)_1	−0.12	0.46	0.28	0.09	−0.02	0.92
LYMP (×10^3^/µL)_1	0.39 *	0.02	0.18	0.27	−0.15	0.38
MONO (×10^3^/µL)_1	−0.07	0.68	−0.01	0.96	0.02	0.90
RBC (×10^6^/µL)_1	0.11	0.51	−0.10	0.54	0.14	0.39
HGB (g/dL)_1	0.04	0.83	−0.03	0.84	0.10	0.56
HCT%_1	0.05	0.78	−0.01	0.95	0.08	0.63
MCV (fL)_1	0.14	0.42	0.28	0.08	−0.01	0.97
MCH (pg)_1	0.19	0.24	0.21	0.21	−0.13	0.44
MCHC (g/dL)_1	0.14	0.39	0.04	0.83	0.07	0.68
RDW-SD fL_1	−0.09	0.59	0.12	0.46	0.02	0.91
RDW-CV_1	−0.22	0.19	−0.14	0.39	0.10	0.53
PLT (×10^3^/µL)_1	0	0.98	0.26	0.12	−0.09	0.59
PDW (fL)_1	0	0.98	−0.33 *	0.04	0.10	0.54
MPV (fL)_1	−0.03	0.86	−0.28	0.09	0.25	0.13
WBC (×10^3^/µL)_12	0.16	0.35	0.26	0.12	−0.30	0.08
NEUT (×10^3^/µL)_12	0.08	0.63	0.35 *	0.03	−0.19	0.26
LYMP (×10^3^/µL)_12	0.01	0.97	−0.10	0.57	−0.07	0.71
MONO (×10^3^/µL)_12	−0.09	0.62	0.12	0.48	−0.18	0.30
RBC (×10^6^/µL)_12	−0.01	−0.01	−0.12	0.49	0.06	0.72
HGB (g/dL)_12	0.07	0.67	−0.14	0.41	0.15	0.38
HCT%_12	0.05	0.78	−0.17	0.34	0.11	0.53
MCV (fL)_12	0.19	0.26	0.07	0.69	−0.10	0.58
MCH (pg)_12	−0.01	0.98	−0.14	−0.41	−0.18	0.30
MCHC (g/dL)_12	−0.70	0.68	−0.14	0.43	0.05	0.79
RDW-SD (fL)_12	−0.19	0.92	−0.30	0.86	0.30	0.88
RDW-CV_12	−0.17	0.34	−0.17	0.34	−0.07	0.68
PLT (×10^3^/µL)_12	−0.03	0.88	0.17	0.33	−0.35 *	0.04
PDW(fL)_12	0.09	0.60	−0.17	0.32	0.07	0.67
MPV(fL)_12	0.14	0.42	0.03	0.86	0.11	0.51

*—*p* ≤ 0.05; _1—results during the first week of hospitalization; _12—results three months after admission to the hospital; FT3—free triiodothyronine; FT4—free thyroxin; HCT—hematocrit; HGB—hemoglobin; LYMP—lymphocytes; MCH—mean corpuscular hemoglobin; MCHC—mean corpuscular hemoglobin concentration; MCV—mean corpuscular volume; MONO—monocytes; MPV—mean platelet volume; NEUT—neutrophils; *p*—*p*-value; PDW—platelet distribution width; PLT—platelets; r_s_—Spearman correlation; RBC—red blood cells; RDW—red blood cell distribution width; TSH—thyrotropin; WBC—white blood cells.

**Table 4 jpm-12-00247-t004:** Relationship of thyroid function assessment parameters with biochemical, routine specialized parameters and pharmacotherapy.

Variable	TSH		FT4		FT3	
	r_s_	*p*	r_s_	*p*	r_s_	*p*
Cholesterol µmol/L_1	−0.05	0.75	0.01	0.97	−0.04	0.81
HDL µmol/L_1	−0.06	0.73	0	0.98	−0.13	0.45
LDL µmol/L_1	−0.12	0.49	−0.06	0.71	0.02	0.88
Triglycerides µmol/L_1	−0.07	0.66	0.08	0.63	0.04	0.81
CRP mg/L_1	0.06	0.74	0.13	0.44	−0.27	0.10
FRAP µmol/L_1	0.02	0.89	−0.10	0.56	0.13	0.44
MDA µmol/L_1	0.07	0.67	−0.25	0.13	−0.07	0.67
PON1 U/L_1	−0.20	0.22	−0.10	0.53	0.09	0.57
FRAP saliva µmol/L_1	0.15	0.36	0.05	0.77	0.39 *	0.02
AVG	0.11	0.56	0.32	0.08	−0.28	0.12
Cholesterol µmol/L_12	−0.13	0.45	−0.04	0.83	−0.12	0.47
HDL µmol/L_12	−0.21	0.21	−0.11	0.52	0.27	0.12
LDL µmol/L_12	−0.18	0.3	0.19	0.27	0.26	0.12
Triglycerides µmol/L _12	−0.25	0.14	−0.18	0.30	−0.32	0.06
CRP mg/L_12	0.01	0.96	0.21	0.22	−0.12	0.50
FRAP µmol/L_12	0.17	0.3	−0.20	0.23	0.21	0.21
MDA µmol/L_12	−0.11	0.5	0.04	0.80	−0.41	0.01
PON1 U/L_12	−0.12	0.47	−0.02	0.92	0.36	0.03
FRAP saliva µmol/L_12	0.19	0.26	0.14	0.41	0.21	0.20
Chlorpromazine sum_1	−0.263	0.110	−0.249	0.099	−0.023	0.111
Chlorpromazine sum_12	−0.131	0.433	0.132	0.556	0.892	0.508

*—*p* ≤ 0.05; _1—results during the first week of hospitalization; _12—results three months after admission to the hospital; AVG—average; CRP—C-reactive protein; FRAP—Ferric reducing ability of plasma; FT3—free triiodothyronine; FT4—free thyroxin; HDL—high-density lipoprotein; LDL—low-density lipoprotein; MDA—malondialdehyde; *p*—*p*-value; PON1—paraoxonase-1; chlorpromazinesum_1, chlorpromazine_sum12 (neuroleptics used at the first and second time point converted to chlorpromazine equivalents). r_s_—Spearman correlation; TSH—thyrotropin.

**Table 5 jpm-12-00247-t005:** Relationship of the thyroid function assessment parameters with the results of specialized imaging examinations.

Variable	TSH		FT4		FT3	
	r_s_	*p*	r_s_	*p*	r_s_	*p*
FA right frontal lobe AVG	0.07	0.7	0.05	0.80	0.07	0.68
FA right frontal lobe DEV	−0.39 *	0.02	0.13	0.47	0.27	0.12
FA left frontal lobe AVG	0.09	0.61	−0.08	0.65	0.06	0.74
FA left frontal lobe DEV	−0.17	0.34	0.06	0.74	0.22	0.22
FA_ACC right AVG	0.02	0.91	0.10	0.59	0.14	0.43
FA_ACC right DEV	0.30	0.08	0.20	0.26	0.24	0.18
FA_ACC left AVG	0.24	0.19	−0.08	0.67	−0.36 *	0.04
FA_ACC left DEV	0.08	0.64	−0.12	0.51	−0.20	0.26
ACC_LIPCR	0.03	0.9	−0.02	0.90	0.23	0.25
ACC_LACCR	−0.05	0.79	0.01	0.97	0.08	0.69
ACC_ALACR	0.09	0.66	−0.02	0.91	−0.03	0.89
ACC_NAACR	0.10	0.6	0.14	0.50	−0.11	0.58
ACC_GLUCR	−0.12	0.54	0.13	0.51	0.44 *	0.02
ACC_GABACR	−0.16	0.44	−0.08	0.68	0.19	0.35
ACC_GLNCR	0.28	0.16	−0.07	0.74	−0.13	0.52
ACC_CHOCR	0.22	0.26	0.10	0.61	0.25	0.20
ACC_GLCCR	0.32	0.1	−0.06	0.76	−0.37	0.06
ACC_MICR	−0.10	0.62	0.48 *	0.01	0.48 *	0.01
ACC_GLUGLNGSHCR	−0.21	0.29	0.24	0.23	−0.23	0.25
ACC_GLCCR_A	0.12	0.54	−0.14	0.49	0.27	0.17
ACC_MICR_A	−0.30	0.12	−0.18	0.37	−0.07	0.73

*—*p* ≤ 0.05; ACC—anterior cingulate cortex; ALA—alanine; AVG—average; CHO—choline; CR—creatine; DEV—standard deviation; FA—fractional anisotropy; FT3—free triiodothyronine; FT4—free thyroxin; GABA –γ-aminobutyric acid; GLC—glucose; GLN—glutamine; GLU—glutamate; GLUGLNGSH—GLU + GLN + GSH complex; LAC—lactates; LIP—lipids; MI—myoinositol; NAA—N-acetyl-aspartate; *p*—*p*-value; r_s_—Spearman correlation; TSH—thyrotropin.

**Table 6 jpm-12-00247-t006:** Descriptive statistics on the analyzed variables in separate groups of patients.

Value	1 (Mean ± SD)	2 (Mean ± SD)
DUP	19.7 ± 19.92	32.47 ± 35.17
Number of hospitalizations (QH)	64.95 ± 31.07	63.88 ± 37.26
Age of onset (AE)	18.6 ± 3.07	19 ± 4.61
Number of episodes (QE)	3 ± 3.4	3.41 ± 3.97
Hospitalization length (HL)	3.49 ± 5.93	3.33 ± 4.79
PANSS pos_1	21.8 ± 4.73	19.71 ± 5.02
PANSS neg_1	25.95 ± 6.03	26.65 ± 7.23
PANSS dis_1	21 ± 5.53	19 ± 5.2
PANSS exc_1	12.25 ± 4.92	13.65 ± 4.57
PANSS emo_1	12.2 ± 3.49	13.29 ± 2.64
PANSS pos_12	9 ± 2.22	8.94 ± 2.62
PANSS neg_12	19.25 ± 4.88	15.81 ± 5.27
PANSS dis_12	10.75 ± 3.32	9.13 ± 2.96
PANSS exc_12	5.85 ± 3.05	5.81 ± 1.68
PANSS emo_12	8.55 ± 2.76	8.13 ± 1.41
BDI_1	11.4 ± 4.19	8.94 ± 4.56
BDI_12	17.95 ± 5.12	13.18 ± 3.13
Calgary_1	4.25 ± 2.17	3.24 ± 1.99
Calgary_12	9.6 ± 2.68	7.29 ± 3
CTQ_EN	18.1 ± 3.92	15.76 ± 5.18
CTQ_EA	14.85 ± 3.88	13.12 ± 5.01
CTQ_PN	13.1 ± 4.73	10.47 ± 4.17
CTQ_PA	9.55 ± 3.36	9.06 ± 4.49
CTQ_SA	6.5 ± 3.5	5.94 ± 3.4
CTQ_TOTAL	62,08 ± 3.45	54,32 ± 3.21
FT4_1	12.78 ± 3.68	12.68 ± 1.31
FT3_1	5.09 ± 1.07	4.6 ± 0.99
Drug dose1_1	12.55 ± 44.25	2.41 ± 2.64
Chlorpromazine1_1	100.83 ± 82.8	76.47 ± 73.14
Drug dose 1_12	69.85 ± 112.13	101.65 ± 147.92
Chlorpromazine 1_12	360.09 ± 86.82	327.45 ± 116.21
WBC × 10^3^/µL_1	5.83 ± 1.4	6.36 ± 3.5
NEUT × 10^3^/µL_1	3.34 ± 1.16	3.79 ± 2.79
LYMPH × 10^3^/µL_1	1.95 ± 0.54	2.36 ± 0.59
MONO × 10^3^/µL_1	0.56 ± 0.18	0.58 ± 0.23
RBC × 10^6^/µL_1	4.72 ± 0.59	4.81 ± 0.64
HGB g/dL_1	13.61 ± 2.2	14.16 ± 1.69
HCT%_1	40.11 ± 5.24	41.99 ± 3.27
MCV fL_1	85.04 ± 4.15	85.52 ± 6.17
MCH pg_1	28.78 ± 2.02	29.67 ± 1.78
MCHC g/dL_1	33.63 ± 1.29	33.99 ± 1.14
RDW-SD fL_1	40.01 ± 3.04	39.57 ± 4.86
RDW-CV_1	13.02 ± 1.22	12.72 ± 1.6
PLT × 10^3^/µL_1	247.15 ± 48.5	236.94 ± 83.91
PDW fL_1	12 ± 2.1	12.2 ± 1.69
MPV fL_1	10.36 ± 0.98	10.5 ± 1.01
PLCR_1	27.58 ± 8.2	28.03 ± 6.25
PCT_1	0.26 ± 0.04	0.25 ± 0.08

(1) 0–2.5 mU/L; (2) >2.5 mU/L; _1—results during the first week of hospitalization; _12—results three months after admission to the hospital; BDI—Beck Depression Inventory; Calgary—Calgary Depression Scale for Schizophrenia; CTQ—Childhood Trauma Questionnaire; CTQ_EA—emotional abuse; CTQ_EN—emotional neglect; CTQ_PA—physical abuse; CTQ_PN—physical neglect; CTQ_SA—sexual abuse; CTQ_TOTAL—total trauma; DUP—duration of untreated psychosis; FT3—free triiodothyronine; FT4—free thyroxin; HCT—hematocrit; HGB—hemoglobin; LYMP—lymphocytes; MCH—mean corpuscular hemoglobin; MCHC—mean corpuscular hemoglobin concentration; MCV—mean corpuscular volume; MONO—monocytes; MPV—mean platelet volume; NEUT—neutrophils; PANSS—Positive and Negative Syndrome Scale; PANSS pos—positive symptoms; PANSS neg—negative symptoms; PANSS dis—disorganized thoughts; PANSS exc—uncontrolled hostility/excitement; PANSS emo—anxiety/depression; PCT—platelet count; PDW—platelet distribution width; PLCR—platelet larger cell ratio; PLT—platelets; RBC—red blood cells; RDW—red blood cell distribution width; WBC—white blood cells.

**Table 7 jpm-12-00247-t007:** Descriptive statistics of the analyzed variables in separate groups of patients.

Variable	1 (Mean ± SD)	2 (Mean ± SD)
Cholesterol µmol/L_1	4.42 ± 0.76	4.62 ± 1.02
HDL µmol/L_1	1.43 ± 0.42	1.29 ± 0.37
LDL µmol/L_1	2.59 ± 0.74	2.54 ± 0.79
Triglycerides µmol/L_1	1.26 ± 0.64	1.39 ± 1.11
CRP mg/L_1	4.2 ± 4.26	7.64 ± 12.13
WBC × 10^3^/µL_12	8.73 ± 2.37	9.42 ± 3.1
NEUT × 10^3^/µL_12	5.22 ± 1.68	5.6 ± 2.47
LYMPH × 10^3^/µL_12	2.04 ± 0.54	2.15 ± 0.75
MONO × 10^3^/µL_12	0.74 ± 0.27	1.22 ± 2.03
FRAP µmol/L_1	0.39 ± 0.22	0.41 ± 0.22
MDA µmol/L_1	0.75 ± 0.16	0.78 ± 0.14
PON1 U/L_1	102.08 ± 5.99	102.04 ± 3.3
FRAP saliva_1	0.42 ± 0.25	0.57 ± 0.34
AVG	339.78 ± 40.8	336.86 ± 31.12
DWI left frontal lobe DEV	31.4 ± 6.42	30.04 ± 5.68
DWI_ACC right AVG	429.41 ± 49.67	441.41 ± 45.17
DWI_ACC right DEV	55.29 ± 10.24	52.12 ± 6.98
DWI_ACC left AVG	419.83 ± 35.17	434.89 ± 50.5
DWI_ACC left DEV	54.97 ± 9.85	52.15 ± 9.92
FA right frontal lobe AVG	0.39 ± 0.04	0.39 ± 0.04
FA right frontal lobe DEV	0.11 ± 0.01	0.1 ± 0.02
FA left frontal lobe AVG	0.36 ± 0.04	0.35 ± 0.1
FA left frontal lobe DEV	0.1 ± 0.03	0.1 ± 0.02
FA_ACC right AVG	0.21 ± 0.05	0.22 ± 0.04
FA_ACC right DEV	0.11 ± 0.04	0.13 ± 0.03
FA_ACC left AVG	0.19 ± 0.03	0.2 ± 0.04
FA_ACC left DEV	0.13 ± 0.13	0.11 ± 0.02
ACC_MI	9,539,129 ± 2,765,434	8,952,829 ± 3,612,834
ACC_MICR	0.77 ± 0.18	0.69 ± 0.17
ACC_MICR_A	0.37 ± 0.33	0.2 ± 0.13

(1) 0–2.5 mU/L; (2) >2.5 mU/L; _1—results during the first week of hospitalization; _12—results three months after admission to the hospital; ACC—anterior cingulate cortex; AVG—average; CR—creatine; CRP—C-reactive protein; DEV—standard deviation; DWI—diffusion-weighted imaging; FA—fractional anisotropy; FRAP—Ferric reducing the ability of plasma; HDL—high-density lipoprotein; LDL—low-density lipoprotein; LYMP—lymphocytes; MDA—malondialdehyde; MI—myoinositol; MONO—monocytes; NEUT—neutrophils; PON1—paraoxonase-1; WBC—white blood cells.

**Table 8 jpm-12-00247-t008:** Results of the statistical test concerning the differences in the scope of the analyzed variables, i.e., in the groups of people divided according to the functioning of the thyroid gland.

Variable	Result of the Statistical Test	
	U	*p*
DUP	127.5	0.19
Number of hospitalizations (QH)	159	0.74
Age of onset (AE)	160	0.76
Number of episodes (QE)	155	0.61
Hospitalization length (HL)	169	0.98
PANSS pos_1	130.5	0.23
PANSS neg_1	159.5	0.75
PANSS dis_1	134.5	0.28
PANSS exc_1	139.5	0.35
PANSS emo_1	135.5	0.29
PANSS pos_12	156.5	0.91
PANSS neg_12	82.5	0.01 **
PANSS dis_12	110	0.11
PANSS exc_12	131.5	0.35
PANSS emo_12	142	0.56
BDI_1	122	0.14
BDI_12	68.5	0 **
Calgary_1	122.5	0.14
Calgary_12	101	0.03 *
CTQ_EN	127.5	0.19
CTQ_EA	135.5	0.29
CTQ_PN	112.5	0.08
CTQ_PA	134.5	0.27
CTQ_SA	132.5	0.13
FT4_1	159.5	0.75
FT3_1	124.5	0.17
Chlorpromazine1_1	140	0.34
Chlorpromazine1_12	138	0.29
WBC × 10^3^/µL_1	162	0.81
NEUT × 10^3^/µL_1	158.5	0.73
LYMPH × 10^3^/µL_1	98.5	0.03 *
MONO × 10^3^/µL _1	169	0.98
RBC 106 uL_1	143	0.41
HGB g/dL_1	151	0.56
HCT_1	143	0.41
MCV fL_1	144	0.43
MCH pg_1	119.5	0.12
MCHC g/dL_1	139.5	0.35
RDW-SD fL_1	148.5	0.51
RDW-CV_1	131.5	0.24
PLT × 10^3^/µL_1	154.5	0.64
PDW fL_1	150	0.54
MPV fL_1	155	0.65
P-LCR_1	157	0.69
PCT_1	141	0.37
WBC × 10^3^/µL_12	141	0.55
NEUT × 10^3^/µL_12	151	0.77
LYMPH × 10^3^/µL_12	158	0.95
MONO × 10^3^/µL_12	157.5	0.94
FRAP saliva 1	118.5	0.12
AVG	107	0.61
FA right frontal lobe AVG	122.5	0.70
FA right frontal lobe DEV	79.5	0.05
FA left frontal lobe AVG	126.5	0.81
FA left frontal lobe DEV	95	0.17
FA_ACC right AVG	111.5	0.43
FA_ACC right DEV	94	0.16
FA_ACC left AVG	108.5	0.37
FA_ACC left DEV	105.5	0.32

*—*p* ≤ 0.05; **—*p* ≤ 0.01;_1—results during the first week of hospitalization; _12—results three months after admission to the hospital; ACC—anterior cingulate cortex; AVG—average; BDI—Beck Depression Inventory; Calgary—Calgary Depression Scale for Schizophrenia; CTQ—Childhood Trauma Questionnaire; CTQ_EA—emotional abuse; CTQ_EN—emotional neglect; CTQ_PA—physical abuse; CTQ_PN—physical neglect; CTQ_SA—sexual abuse; CTQ_TOTAL—total trauma; DEV—standard deviation; FA—fractional anisotropy; FT3—free triiodothyronine; FT4—free thyroxin; FRAP—ferric reducing ability of plasma; HCT—hematocrit; HGB—hemoglobin; LYMP—lymphocytes; MCH—mean corpuscular hemoglobin; MCHC—mean corpuscular hemoglobin concentration; MCV—mean corpuscular volume; MONO—monocytes; MPV—mean platelet volume; NEUT—neutrophils; *p*—*p*-value; PANSS—Positive and Negative Syndrome Scale; PANSS pos—positive symptoms; PANSS neg—negative symptoms; PANSS dis—disorganized thoughts; PANSS exc—uncontrolled hostility/excitement; PANSS emo—anxiety/depression; PCT—platelet count; PDW—platelet distribution width; PLCR—platelet larger cell ratio; PLT—platelets; RBC—red blood cells; RDW—red blood cell distribution width; U—U-Mann–Whitney test; WBC—white blood cells.

**Table 9 jpm-12-00247-t009:** Descriptive statistics on the cluster analysis.

	TSH Level		TSH Level	
	1–2.5 mU/L	2.51–4.2 mU/L	Cluster 1	Cluster 2
n	20	17	20	17
%	54.1	45.9	54.1	45.9

**Table 10 jpm-12-00247-t010:** Correlations between chlorpromazine equivalents and thyroid function markers.

Spearman Correlations		Chlorpromazine sum_1	Chlorpromazine sum_12
TSH_1	Correlation coefficient	−0.263	−0.131
	Significance (bilateral)	0.110	0.433
FT4_1	Correlation coefficient	−0.249	0.099
	Significance (bilateral)	0.132	0.556
FT3_1	Correlation coefficient	−0.023	0.111
	Significance (bilateral)	0.892	0.508
Cortisol μg/dL_1	Correlation coefficient	0.081	−0.047
	Significance (bilateral)	0.623	0.777
CRP mg/L_12	Correlation coefficient	0.183	0.291
	Significance (bilateral)	0.277	0.080

## Data Availability

The data are available from the corresponding authors upon reasonable request.

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
