# Peer review of "Redefining the Cut-Off Ranges for TSH Based on the Clinical Picture, Results of Neuroimaging and Laboratory Tests in Unsupervised Cluster Analysis as Individualized Diagnosis of Early Schizophrenia"

_jpm, 2022, doi:10.3390/jpm12020247_

Round 1

Reviewer 1 Report

Congratulations on this work!

I have some remarks:

I.There is evidence that inflammation may play a role in schizophrenia, oxidative stress being involved in the pathogenesis of this psychiatric disease. C-reactive protein is not only widely used as a biomarker of inflammatory status, but CRP levels are positively correlated with schizophrenia positive symptoms.

Regarding this, I have some questions:

1.Why acute inflammatory diseases and the use of glucocorticoids are not among your exclusion criteria?

2.What is the relationship between the positive symptoms and the measured CRP level? Does TSH correlate with CRP in your research, respectively?

You can find a previous study (https://doi.org/10.2147/NDT.S322005) in which correlation between CRP level and positive symptoms of schizophrenia was described, and about the role of steroid inflammatory drugs on TSH level( DOI 10.1007/s40618-014-0133-2), respectively.

II.Second-generation antipsychotics differ from each other by the partial D2-agonist effect (aripiprazole, brexpiprazole, cariprazine). Stimulating D2 receptors, aripiprazole and similar SGAs can downregulate TSH and PRL secretion levels. After a case report ( (https://doi.org/10.1136/bcr-2017-220192 ), low plasma TSH and PRL levels were recommended to be used as simple biological markers to assess the side effects of aripiprazole in schizophrenia patients.

Regarding this, I have some recommendations:

1.Specify if you included patients treated with partial D2 agonists

2.In statistical calculations to separate patients treated with D2 antagonists and partial D2 agonists to clarify TSH response on the different types of antipsychotics.

Author Response

                                                                                                                                     February 5th, 2022             

Letter to the Editor

Dear Editor,

Please find attached a corrected version of our paper "Redefining the cut-off ranges for TSH based on the clinical picture, results of neuroimaging and laboratory tests in unsupervised cluster analysis as individualized diagnosis of early schizophrenia“, an original paper by Åšmierciak et al. We would like to thank the Reviewer for their generally positive assessment of our manuscript and for their helpful comments. We have tried to follow all of them and we have been able to fulfill almost all the Reviewers’ requirements. We believe this has allowed us to improve the quality of the paper.

We would be grateful if You consider our revised manuscript for publication in the Journal of Personalized Medicine. In any case, we would like to let You know that we are open to consideration of any further comments on our answers.

Dear Reviewer,

Thank You very much for the positive evaluation of our manuscript and helpful suggestions. Please find the responses to each of the suggestions below:

https://docs.google.com/document/d/1CITA7N_gCo2uvub0vyIEma-7iTv9fhK_/edit?usp=sharing&ouid=105568337571193698272&rtpof=true&sd=true

Sincerely,

Authors

Reviewer 2 Report

Åšmierciak et al. present an interesting study which demonstrates thyroid hormones to be associated with clinical features, trauma history, inflammatory markers, and neurometabolites among inpatients with schizophrenia. I liked the multimodal approach taken by authors and think that this study provides some useful insights to the field. However, there are a number of issues which need to be addressed before I can recommend this article for publication. Detailed comments are appended below.

  1. Could authors please define PN in the same way that they define the other types of trauma in their methods section.
  2. What I presume is a typo on line 231 (authors refer to Ff3 rather than FT3).
  3. Authors describe their cluster analysis in the following way:

“An unsupervised cluster analysis was used to assess the relationship between all recorded TSH values and the clinical and laboratory assessment of patients in early psychosis. Using the k-means cluster analysis, 2 clusters of people were distinguished, i.e. with the maximum possible difference in the level of TSH”

I have couple of questions/comments about this. Firstly, can the authors be more specific about exactly which variables were entered into the cluster analysis? As it has been written, I’m left wondering whether everything was entered in the k-means cluster analysis (all clinical assessments, trauma measures, laboratory assessments, TSH, FT3, FT4, etc?) or whether it was just THS with the intention being to explore the relationship between the derived clusters and the clinical and laboratory assessment following the cluster analysis?  Second, if everything was entered, did the authors appropriately scale all of the variables entered into the clustering analysis? Finally, the k-means approach works by minimising within-cluster variance rather than by maximising inter-cluster “difference” as the authors stated.

  1. Dozens of statistical association tests and comparisons performed, yet no correction for multiple testing. This increases the likelihood that authors may have made some type I errors and should be noted as a limitation in the discussion.
  2. Table 2a is a bit hard to read. The “ACC_MI” row in particular is too messy and it’s not entirely clear where one column starts and the next begins. The label “2a” is also applied to two separate tables when the second one should be 2b. Could authors consider replacing table 2a/b with a series of histograms, stem and whisker plots, or density plots (all plotted on a grid)? I think such figures would make it much easier for the reader to interpret all of the information that is presented in the tables.
  3. Authors noted that “The selected clusters show identical descriptive statistics of the analyzed variables, as in the division into groups of people based on the TSH level”. Could the authors please confirm that not only were the descriptive statistics the same, but that these groups actually were the same? The unsupervised clustering could theoretically have produced two groups of the same size as the groups derived based on the TSH level, but with a different cut-off point.
  4. I’m not sure how the ROC curve was calculated and by extension, I’m not convinced of its relevance. Was the ROC curve showing the accuracy of the TSH levels at 1 week post hospitalisation in predicting BDI levels at 12 weeks post hospitalisation? If so, then how was BDI dichotomised to allow for the calculation of sensitivity and specificity? If not, and instead the ROC curve was showing the predictive utility of BDI at 12 weeks in predicting TSH levels at 1 week, then I don’t really see the point of the ROC curve at all. ROC curves are generally used to demonstrate the accuracy of a particular test/measure in predicting some outcome. However, I can’t imagine why you would want to use depression symptoms at 12 weeks post hospital admission to predict TSH levels at 1 week post hospital admission? I can see the potential utility of doing the opposite (using TSH levels to predict depression levels in the future). However, a validation plot would be more appropriate for evaluating the performance of TSH on BDI given that BDI is a continuous outcome.
  5. To be of any clinical utility, THS would need to show adequate accuracy in predicting the course of illness. A statistically significant association is no guarantee of this, so authors should demonstrate the accuracy of THS in predicting clinical progression through validation plots.
  6. The following statements in the discussion need to either be reworded to make them sound less definitive (e.g. use words like “theorised” rather than “assumed”) or to be backed up by specific citations as they are quite substantial claims to make without adequate evidence to back them up:
    1. “Oral metabolism disorders often precede the actual onset of schizophrenia and increase with the development of the disease. Very often, these changes are associated with dysbiosis of the salivary microflora, which is the causative factor of the onset of schizophrenia.”
    2. “This is also supported by the second negative correlation between TSH and the level of myoinositol in the ACC, which can be additionally explained by the swelling of the examined areas of the brain tissue (although the measurements of thickness or gray to white matter ratios were not performed), which is caused by a slow flow of components in the analyzed brain regions.”
    3. “it can be assumed that the negative correlation between TSH and DEV is related to the limited movement of micronutrient particles from astrocytes to neurons within the right frontal lobe”.
    4. “The dependencies between TSH and myoinositol in ACC shown by us, according to the literature, indicate that in this area of the brain, astrocyte cells are damaged”
  7. I don’t think that the following statement is justified by the evidence available:

“Correlation between FT4 and MI/CR in this study proves that the increase in myoinositol, which is a marker of glial cells representing the membrane transport of nutrients for neurons, reflects glial activation in patients with psychosis”.

I fail to see how a correlation between free thyroxine and myo-inositol creatine gives any evidence (let alone proof) that increased myo-inositol reflects glial activation in psychosis. It’s not as if FT4 levels are a gold standard test for glial activation upon which the authors could substantiate the claim that increases in myo-inositol reflect glial activation based on the fact that it was associated with FT4.

  1. Could the authors please comment on the generalisability of their findings given the extensive exclusion criteria implemented? Many people who present with psychotic disorders (e.g. people who smoke, have comorbid cardio vascular diseases, etc.) were excluded from this study. How useful of a biomarker would TSH be for people who didn’t meet the exclusion criteria of this study?
  2. One of my biggest concerns with this study is the potential for findings to have been confounded by the different types/dosages of psychotropic drugs patients received during their treatment. This concern is further exacerbated by the fact that that there were apparent associations between clinical symptoms and thyroid hormones at baseline (1 week post hospitalisation). This is because the symptoms patients presented with may have influenced their treatment, thus the associations evident at 12 weeks may be attributable to the baseline levels of TSH, FT3, and FT4 as the authors suggest; or they may be attributable to the fact that people with different symptoms at baseline received different treatment. Is there any way for the authors to account for medication in their analyses, or failing that, to discuss this issue in their limitations section?

Author Response

                                                                                                                             February 6th, 2022                    

Letter to the Editor

Dear Editor,

Please find attached a corrected version of our paper "Redefining the cut-off ranges for TSH based on the clinical picture, results of neuroimaging and laboratory tests in unsupervised cluster analysis as individualized diagnosis of early schizophrenia. “, an original paper by Åšmierciak et al. We would like to thank the Reviewer for their generally positive assessment of our manuscript and for their helpful comments. We have tried to follow all of them and we have been able to fulfill almost all the Reviewers’ requirements. We believe this has allowed us to improve the quality of the paper.

We would be grateful if You consider our revised manuscript for publication in the Journal of Personalized Medicine. In any case, we would like to let You know that we are open to consideration of any further comments on our answers.

Dear Reviewer,

Thank You very much for the positive evaluation of our manuscript and helpful suggestions. Please find the responses to each of the suggestions below:

https://docs.google.com/document/d/1gUnJfZMKJ66Q6ULhofQDRUpq4DPNT9I3/edit?usp=sharing&ouid=105568337571193698272&rtpof=true&sd=true

Sincerely,

Authors
